# Frequent disturbances enhanced the resilience of past human populations

Philip Riris[1✉], Fabio Silva[1], Enrico Crema[2], Alessio Palmisano[3], Erick Robinson[4,5,6], Peter E. Siegel[7], Jennifer C. French[8], Erlend Kirkeng Jørgensen[9], Shira Yoshi Maezumi[10], Steinar Solheim[11], Jennifer Bates[12], Benjamin Davies[13], Yongje Oh[12] & Xiaolin Ren[14]

The record of past human adaptations provides crucial lessons for guiding responses to crises in the future[1–3]. To date, there have been no systematic global comparisons of humans' ability to absorb and recover from disturbances through time[4,5]. Here we synthesized resilience across a broad sample of prehistoric population time–frequency data, spanning 30,000 years of human history. Cross-sectional and longitudinal analyses of population decline show that frequent disturbances enhance a population's capacity to resist and recover from later downturns. Land-use patterns are important mediators of the strength of this positive association: farming and herding societies are more vulnerable but also more resilient overall. The results show that important trade-offs exist when adopting new or alternative land-use strategies.

Understanding the range of past responses of human societies to disturbances is a global priority across the social and natural sciences and will support the development of solutions to future crises[1–3]. Numerous case studies have addressed past cultural collapse, transformation and persistence, although how to best characterize these processes is a subject for debate[6]. A major unresolved issue is the lack of comparability between cases of population resilience in the historical sciences[4,5]. Few studies explicitly model impacts, recovery and adaptation, or formally account for long-term history, which contains important and previously overlooked variation within and between cultural or environmental settings. Furthermore, a tendency to narrowly focus on responses to extreme events in both natural and social systems[7,8] may overemphasize local or short-term adaptive success at the expense of understanding large-scale or long-term vulnerabilities[6,9]. A well-known example is the shift to a narrow marine diet among the Greenland Norse that initially offset the short-term risk of crop failure yet heightened societal vulnerability to longer-term North Atlantic cooling[10]. Here, we establish a global comparative approach to long-term resilience to identify the factors that structure the response of prehistoric populations to disturbances through time. The approach measures population capacity to withstand changes, as well as the rate of recovery following a disturbance through the common proxy of radiocarbon time–frequency data[11,12]. Disturbances are the inferred drivers of relative reductions in population or archaeological activity in prehistory, which are described variously as recessions, downturns, busts, negative deviations or similar[12–15] and form the focus of this study, using summed probability distributions (SPDs) of calibrated radiocarbon dates. SPDs function as an index of relative levels of human activity, or population change, over time[16,17]. Population downturns are defined as periods when SPDs are significantly below expected growth trajectories

in response to disturbances. Our efforts focus on two key questions: (1) how quickly do past populations recover after downturns; and (2) what factors mediate past resistance and resilience to downturns?.

To quantify patterns in population resistance–resilience, we performed a meta-analysis of 16 published study regions that used archaeological radiocarbon data to reconstruct regional palaeodemographic trends (Supplementary Table 1). Our approach trades specificity for a large-scale, comparative perspective that still accounts for variation between cases to focus on the emergent properties of the statistical analysis. Studies were reviewed based on three criteria: evidence for significant downturns, their scope and the inclusion of radiocarbon datasets. A lack of any single criterion resulted in the exclusion of a study. Cases with no reported downturns were not included, nor were those whose scope was restricted to specific activities within a regional radiocarbon dataset, such as flint mining[18]. Where published data have been superseded by later compilations, dates were added from the People3k database[19], a systematic compilation of cleaned radiocarbon dates, based on the geographical area of the original study. Our global sample of regions ranges from the Arctic to the tropics and spans 30,000 years of history (Fig. 1a). Population downturns were reproduced using our protocol (Methods) and resistance–resilience metrics (Fig. 1b) were collected on 154 periods of population downturn, with a median of 8.5 periods in each region (Fig. 1c and Supplementary Tables 1 and 2). The numerical metrics collectively capture the severity, chronology and frequency of periods of statistically significant population downturn. Disturbances were classified into both general categories and specific drivers according to the original studies and expert interpretation of regional social, cultural and environmental history (Extended Data Table 1). The broad category of land use and evidence for adaptive change during, or in the wake of, downturns were also recorded. The

[1]Department of Archaeology and Anthropology, Bournemouth University, Poole, UK. [2]Department of Archaeology, University of Cambridge, Cambridge, UK. [3]Department of Historical Studies, University of Turin, Torino, Italy. [4]Native Environment Solutions, Boise, ID, USA. [5]Division of Atmospheric Sciences, Desert Research Institute, Reno, NV, USA. [6]School of Human Evolution and Social Change, Arizona State University, Tempe, AZ, USA. [7]Department of Anthropology, Montclair State University, Montclair, NJ, USA. [8]Department of Archaeology, Classics and Egyptology, University of Liverpool, Liverpool, UK. [9]NIKU High North Department, Norwegian Institute for Cultural Heritage Research, Tromsø, Norway. [10]Department of Archaeology, Max Planck Institute of Geoanthropology, Jena, Germany. [11]The Museum of Cultural History, University of Oslo, Oslo, Norway. [12]Department of Archaeology and Art History, Seoul National University, Seoul, Republic of Korea. [13]Environmental Studies, Tufts University, Boston, MA, USA. [14]Institute for the History of Natural Sciences, Chinese Academy of Sciences, Beijing, People's Republic of China. ✉e-mail: priris@bournemouth.ac.uk

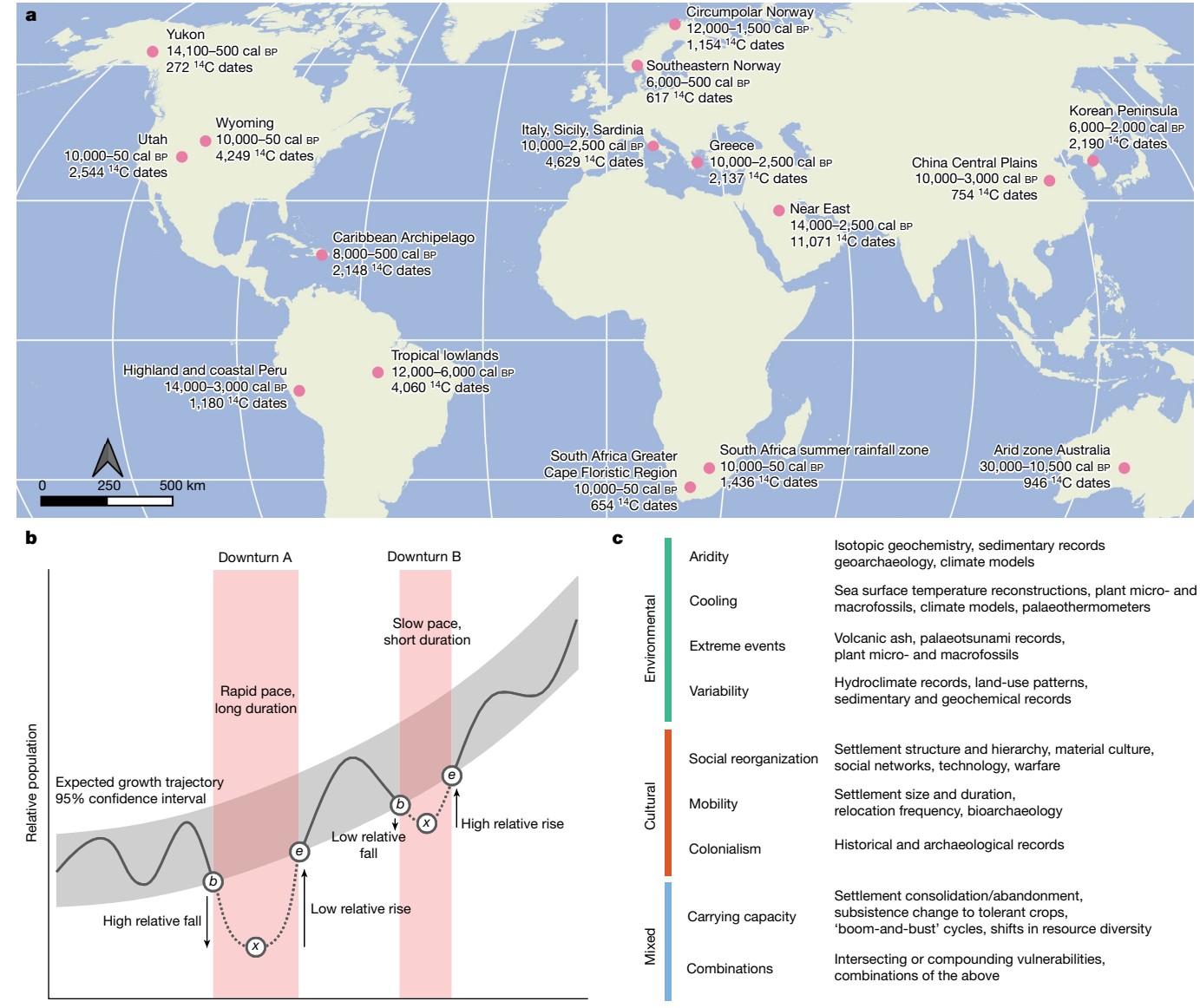

**Fig. 1 | Map and summaries of all world regions and disturbance types included in this study. a**, World map of study regions made with Natural Earth. Time ranges in calendar years before present (cal BP) and [14]C dates used in this meta-analysis are indicated for each region. **b**, Conceptual diagram of measuring resistance–resilience on palaeodemographic downturns against expected growth trajectories. Downturn A is longer and faster, with low resistance and low relative rate of recovery (resilience). Downturn B is shorter and slower, displaying higher resistance and higher resilience. Other combinations of high or low resistance and/or resilience are possible (Extended Data Fig. 4). Parameters *b*, *x* and *e* for the equations in Table 1 are indicated in white dots. **c**, Summary of disturbance types within three broad categories reported in published palaeodemographic studies. Listed proxies are examples drawn from the surveyed literature. 'Unclear' downturns (downturns with a lack of clear evidence for a driver) are not shown (Extended Data Table 1).

focus of this meta-analysis is to identify the factors governing the relative depth of downturns (resistance) and the rate of recovery at their conclusion (resilience). A suite of hierarchical linear mixed models was developed to test for significant associations between parameters while controlling for regional variability (Methods).

This approach to past resistance–resilience provides a glimpse into population-level responses to disturbances throughout human history. Results demonstrate that a single factor—the frequency of downturns—increases both the ability to withstand disturbances and to recover from them. Additionally, the frequency of downturns experienced by a given population is influenced by land use: agricultural and agropastoral populations experience significantly more downturns over time than other land-use patterns recorded during downturns (Extended Data Table 1). These findings collectively suggest that the global shift to food-producing economies during the Holocene (starting 11,700 calendar years before present) may have increased population vulnerability to disturbances, yet in the process enhanced their adaptive capacity through repeated exposure. Parallels to our long-term perspective on human population change in macroecology suggest that comparative approaches in the historical sciences have the potential to generate profound insights into past human–environmental relationships on a broad scale.

## Cross-sectionally high resilience

The most common high-level driver of downturns, after those with a lack of evidence for a specific cause (unclear, *n* = 65), is environmental (*n* = 48, 31%), followed by mixed (cultural–environmental) (*n* = 33, 21%).

## Table 1 | Metrics and model parameters extracted from global case studies of past population downturns

| Name | Description | Notes |
|---|---|---|
| **Parameters** | | |
| Overall duration | Length of downturn, in calendar years before present | |
| $n_{downturn}$ | Cumulative number of downturns in a region | |
| Time to minimum | Time to SPD minimum value, in calendar years before present | |
| $T_{start}$ | Calendar year before present at start of downturn | |
| $T_{end}$ | Calendar year before present at end of downturn | |
| $b$ | Baseline SPD value at the start of a downturn | Fig. 1 and Extended Data Fig. 4 |
| $x$ | Minimum SPD value during a downturn | |
| $e$ | SPD value at the end of a downturn | |
| **Independent** | | |
| Resistance | The depth of a downturn relative to baseline conditions. Range: 0–1 | $1 - \frac{2 \times \| b - x \|}{\| b \| + \| b - x \|}$ |
| Resilience | Rate of recovery to baseline conditions, controlling for maximum impact of downturn. Range: −1–1 | $\frac{2 \times \| b - x \|}{\| b - x \| + \| b - e \|} - 1$ |
| **Dependent** | | |
| Geographical information | Latitude, longitude, world region. | |
| Relative pace | The time to SPD minimum normalized by downturn duration. Range: 0–1 | $\frac{time\ to\ minimum}{overall\ duration}$ |
| Frequency of downturns | Cumulative number of downturns per region standardized by duration in calendar years, per millennium. | $\left( \frac{n_{downturn}}{duration} \right) \times 1,000$ |
| Category | Driver of disturbance. | Extended Data Table 3 |
| Type | Type of disturbance. | |
| Land use | Dominant land-use pattern. | |
| Change | Cultural changes over the interval of the downturn. | Boolean variable, with specific responses noted separately |

Detailed descriptions of resistance–resilience metrics are in Methods.

The most common disturbance type is aridity in the environmental category ($n = 31$), followed by mobility ($n = 20$) in the cultural category. Only a third of recorded downturns have resistance values that drop more than 50% from pre-downturn activity levels ($n = 53$, 34.4%; Fig. 2a). Most downturns ($n = 133$, 86%) end before baseline relative population levels are attained; in other words, observed SPD values are lower at the end of most downturns than at the pre-disturbance reference value. Resilience is relatively high across all cases (median = 0.64, $n = 154$), with 40% ($n = 63$) still attaining 90% of pre-downturn conditions by their end (Fig. 2b). Full returns to SPD baseline conditions are frequently interrupted by subsequent downturns. Downturns associated with cultural drivers return the highest median resilience (0.74), whereas the median mixed (cultural–environmental) resistance is highest at 0.79. Resistance (0.65) and resilience are lowest among climate-driven downturns (0.57). However, we do not find support for significant differences in either metric across disturbance categories (analysis of variance, resistance: $F = 0.541$, $P = 0.65$, d.f. = 3, $n = 154$; resilience: $F = 0.04$, $P = 0.98$, d.f. = 3, $n = 154$).

## Global variation in recovery rate

Initial modelling indicates that the geographical location of downturns does not affect resistance, with the exceptions of significantly higher values in the Caribbean archipelago and the Italian peninsula. Conversely, there is support for significantly lower values for resilience in three regions: the Central China Plains, the Caribbean archipelago and the Korean Peninsula, when compared with all other regions (Extended Data Table 2). Further examination of these cases reveals that a large minority of downturns in these three regions return negative values of resilience ($n = 11$, 39%), which are produced when the population proxy exceeds the SPD value at the start of the downturn by its end (Extended Data Fig. 4, 12). Although the SPD population proxy remains below modelled expectations in all of these cases and therefore they are, in the strictest sense, downturns relative to the null models, the results nevertheless imply that populations in these regions were, on average, able to recover faster than the norm. Owing to the observed range of variation and its potential impact on the metrics, study region was retained as a random effect variable in the mixed-effect models.

## Long-term downturns are the norm

The durations of population downturns tend towards centennial (100–500 years, $n = 48$, 31%) and decadal periods (less than or equal to 50 years, $n = 47$, 30.5%), with a median of 98 years across the sample. Downturns lasting longer than 500 years are a minority ($n = 29$). The time taken to reach SPD minima skews further towards decadal timescales (Fig. 2c). Only a single downturn that commences with the 8.2-thousand-years-ago event in the Near East[20] has a time to minimum longer than a millennium (2,070 years). Both variables have a strong positive skew (duration = 2.84, time to minimum = 4.23). To control for the distribution and broad range of variation in these variables, the time to minimum was normalized by downturn duration to produce an index of its relative speed, which we term 'pace' (Fig. 2d). This transformation enables comparison of the variation between downturns, with higher numbers reflecting slower downturns ($\sigma = 0.55$, s.d. = 0.23) and lacking support for non-normally distributed values as in the time to minimum and duration variables (Shapiro–Wilk $W = 0.98396$, $P = 0.07108$). Consequently, relative pace was employed as a fixed-effect candidate in the mixed-effect modelling.

## Land use mediates resilience

The frequency of downturns over time by region was estimated by normalizing the cumulative number of downturns in a region by their duration (Table 1). We transformed this to the logarithm of events per millennium to account for its strongly skewed distribution. The variable allows us to compare the rate at which downturns occur. It consistently displays the strongest relationship to both resistance and resilience ($P < 0.001$ in both cases) (Extended Data Table 3) and is the only fixed variable retained by the information criterion-based selection procedure.

Collectively, these results indicate that populations experience an enhanced ability to withstand disturbances as frequency increases, as well as to recover in the aftermath (Fig. 3a,b). Further examination of frequency of downturns shows that, among the recorded disturbance types, changes to mobility regimes (median frequency of downturns = 2.26, $n = 20$) and high environmental variability (median frequency of downturns = 2.19, $n = 17$) occur at the highest rate per millennium, whereas cooling occurs at the lowest rate (median frequency of downturns = 0.761, $n = 4$) (Fig. 3c). In terms of regional variation, the highest frequency of downturn is recorded in the South African Greater Cape Floristic Region (median frequency of downturns = 2.41, $n = 17$ over 9,950 years) and the lowest in the Korean Peninsula (median frequency of downturns = 0.58, $n = 3$ over 4,000 years).

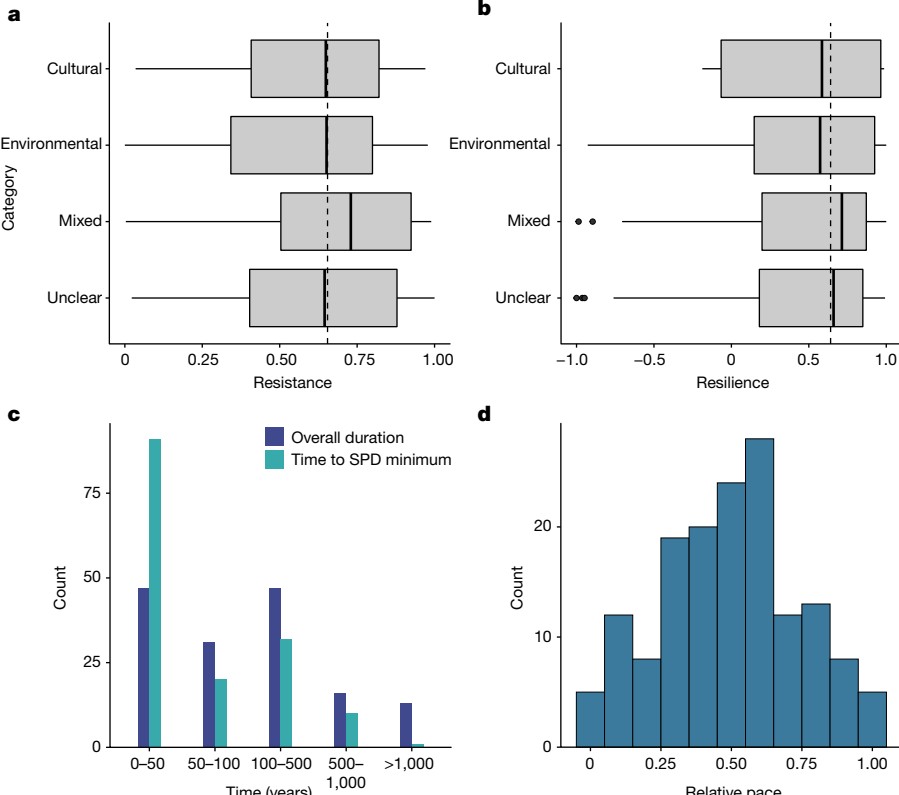

**Fig. 2 | Summaries of downturn numerical metrics showing the relationship between resistance, resilience and the duration and relative pace of downturns. a,b**, Distributions of resistance (**a**) and resilience (**b**) across disturbance categories (Extended Data Table 1). The lower and upper hinges correspond to the 25th and 75th percentiles The upper and lower whiskers extend from the hinges to 1.5 × the interquartile range (IQR). The thick black line represents the group median. Data beyond the whiskers are individually plotted outlying points. The dashed line represents the combined data median. **c**, Distribution of the duration of downturns and the time to SPD minima across all recorded downturns. **d**, Relative pace (overall duration normalized by time to minimum) approximates a normal distribution and enables comparison of the speed of downturns. Modelled downturn durations skew towards multidecadal and centennial timescales.

Treating the frequency of downturns as a response variable (Methods) revealed that agricultural and agropastoralist land-use patterns are associated with significantly higher rates of downturn compared with low-level food production, marine foraging or mixed subsistence. Results from this additional modelling exercise suggest that the frequency of downturns is likely to have an important effect on resistance and recovery, whereas the frequency of downturns itself covaries with the dominant pattern of land use and disturbance type in a given time and place. A larger sample size of downturns would increase the explanatory power of our approach and enhance our characterization of these relationships.

## Discussion

This meta-analysis has examined the potential factors influencing resistance and resilience across a broad archaeological sample, and was controlled for regional variation. The frequency of downturns is the main determinant of the observed outcomes among the examined factors. The relationship between resistance and resilience displays variable rates, although these events all tend to unfold at multidecadal to centennial timescales. Well-known historical examples support this finding: the catastrophic depopulation of indigenous groups across the Americas took place over centuries[21], and the collapse of imperial power in Western Rome was preceded by a long period of rural depopulation[22]. Systematic data on independent population downturns in prehistory are less common. However, what data are available[23,24] corroborate this result: palaeodemographic downturns resolved in radiocarbon data tend to last at least one human generation but frequently much longer.

The frequency of downturns is associated with both the ability of past populations to withstand downturns and the rate of recovery following them across a broad sample of human populations. The results suggest the existence of a common mechanism among human populations that confers resilience to disturbances. The size of this interaction is greater for resistance (eta-squared ($\eta^2$) = 0.46, $P < 0.001$; Fig. 3b) than resilience ($\eta^2$ = 0.29, $P < 0.001$; Fig. 3a). In practical terms, this suggests that the ability to withstand downturns is distinct from the ability to recover in their wake. We note that this result does not imply a monocausal or deterministic relationship; there are likely to be several adaptive pathways that increase population resistance and resilience by means of the mechanism of increasing downturn frequency.

Further testing indicates that land-use practices associated with food production, such as farming and herding, are significantly and positively correlated with the frequency of downturns (Fig. 3c and Extended Data Table 3). From the early Holocene, the proportion of land use associated with food production in our sample of downturns also increased, as the aggregate global population became gradually more reliant on domesticated species for meeting subsistence needs (Fig. 3d). Collectively, these trends show that although populations generally increased resistance and resilience over time, the heightened rate of downturn over time is itself likely to be linked to the historical tendency towards food-producing subsistence systems. Current archaeological evidence does not indicate that this process was unidirectional or inevitable; foraging and food production are neither mutually exclusive nor in opposition. Our synthetic findings agree with specific cases showing that the behavioural and social changes that food production entailed had trade-offs in other arenas[23].

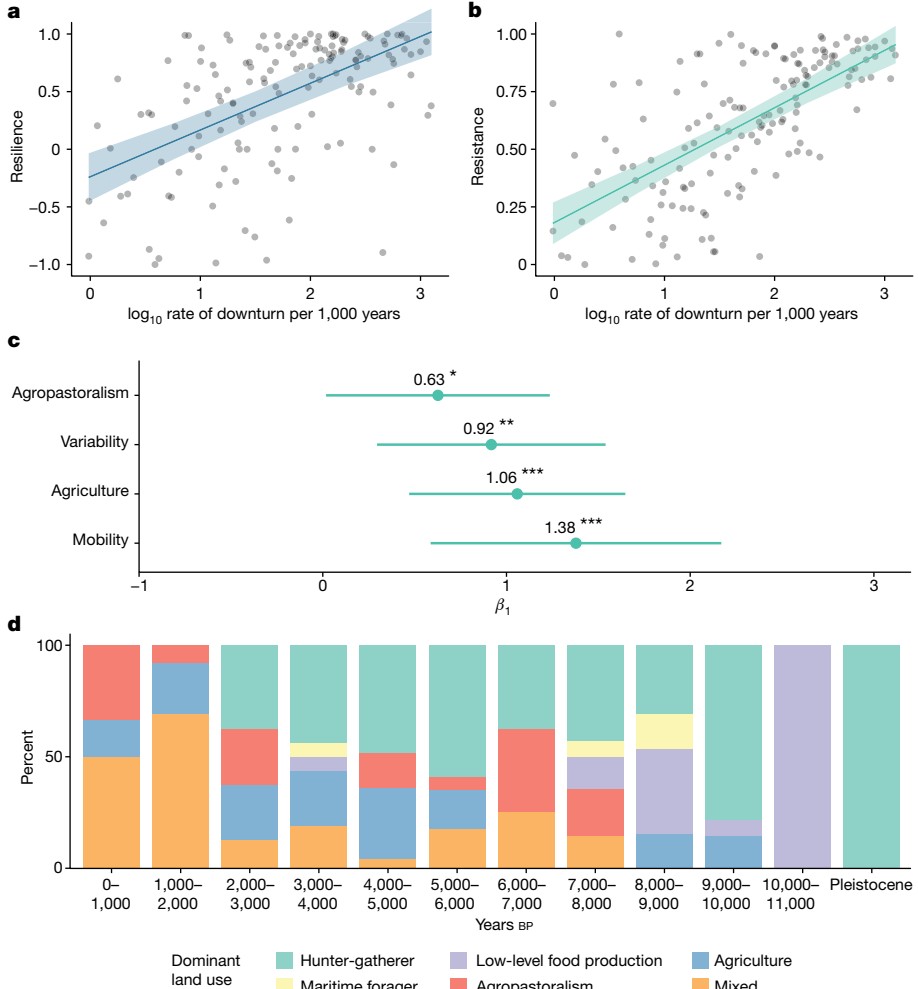

**Fig. 3 | Effect sizes of model terms and dominant land use during downturns over time. a,b**, Resilience (**a**) and resistance (**b**) are strongly influenced by the rate of downturn per millennium, with a stronger effect for resistance. Values are extracted from fitted models I and II. **c**, Standardized regression coefficients for significant (*P* < 0.05) mixed-model terms for *n* = 154 independent samples based on a two-sided test. Hunter-gatherer was set as the reference group for land use. **d**, Proportions of dominant land-use types during downturns, in 1,000-year time slices. Pleistocene downturns (before 11,000 calendar years before present, *n* = 24) have been combined. The 95% confidence intervals are indicated in **a**–**c** by shaded bands and error bars. *\*P* < 0.05, *\*\*P* < 0.01 and *\*\*\*P* < 0.001.

Traditional agricultural or agropastoral practices include a diverse range of socio-ecological systems that are often highlighted as potential sources of inspiration for solutions to current sustainability, biodiversity and conservation challenges[5,10,25,26]. The results suggest that population downturns or collapse are an inherent property of these systems and a potential trade-off of promoting their use. Systematic reviews in disturbance ecology indicate that frequent natural disturbances enhance the long-term resilience of key ecosystem services and that localized subsystem declines are an important mechanism through which this occurs[27]. Our study provides insight into the possible existence of an analogous relationship within our sample of human populations; higher frequencies are strongly correlated with both smaller downturns and closer matches to pre-downturn values in the SPD proxies. We suggest that humanity's overall constant long-term population growth[28] may have been sustained due, in part, to the emergent positive feedback between vulnerability, resistance and recovery documented here.

Population decline has been termed an 'inevitable' feature of our species' demographic dynamics[29]. In their systematic review of historical collapse and resilience dynamics, Cumming and Peterson[1] list depopulation as a key metric or factor in every ancient socio-ecological system studied. We anticipate that this singular status will continue undiminished. Our contribution indicates that downturns play an important role in human population history by enhancing the resilience of survivor populations. We speculate that the creation of biased cultural transmission may be responsible; downturns provide critical opportunities for landscape learning and the strengthening of local-to-regional knowledge networks to propagate through a cultural system[30,31]. Population downturns have been identified as potential triggers of labour investment in infrastructure, social cohesion and technological advancement[15]. Together, these mechanisms have the potential to enhance the preferential transmission of knowledge and practices that promote future resistance or resilience[10]. Raising population thresholds by intensifying land use may also heighten the risk of more serious collapse in return for increasingly marginal benefits[1,24,32]. Other non-trivial and historically contingent factors that are likely to affect outcomes are the diversity and ecology of domesticated species assemblages, degree and type of political complexity, and settlement patterning in relation to the environment. Indicators such as the cessation of monument construction, loss of literacy or economic turmoil can provide additional insight into the consequences and/or potential drivers of population downturns. These potential causal links must be rigorously tested, however, as they are not easily disentangled. A realistic model for the generative mechanism underlying the resilience of human populations will therefore have to be multiscalar and sensitive

to cascading effects to account for how various exogenous impacts unfold and endogenous strategies are developed to solve them. The approach used for our comparative analysis of palaeodemographic resistance–resilience, however, does not distinguish between these elements of the studied populations. Future research may translate between our work and the microscale, the patterns of which are only truly understandable within the kind of generalist, synthetic frame of reference that we provide.

This study finds parallels in macroecology, where analogous resistance–resilience outcomes have been suggested to only fully resolve at centennial timescales or above[33,34]. Archaeology is uniquely suited to examining past population history, and the dynamics that underlie these trends, at such long-term timescales[35–37]. Understanding past societies' responses to crises is often explicitly motivated by the goal of applying learnings from the historical sciences to present-day policy and activism, contributing to the ultimate objective of fostering resilient adaptations for the future[6]. Most archaeological work on past resilience is historically particularistic[4,9] and emphasizes the contingencies, decisions and practices that underwrote successful adaptations in specific times and places[38,39]. This specificity can be illuminating but, if the historical sciences are to play any role in fostering future resilience, improving our understanding of the processes and drivers that influence long-term, centennial-scale resilience is a necessary prerequisite[5]. Our approach has highlighted the global relationship between population change and frequency of disturbance over millennial timescales and applies across a broad geographical and chronological sample of past populations, including prehistoric examples that have been overlooked in systematic reviews of societal resilience more broadly. Improved clarity on the drivers of exposure frequency and type in the past will help to reveal the mechanism(s) behind the dynamics we describe and their potential limits, which is particularly important as environmental variability is predicted to increase in the future[40,41]. Archaeological and historical case studies have focused on the frequency of volcanism, warfare[42] and hydroclimatic oscillations[23,43], as well as the rate of abrupt or extreme events in general[8]. Comparable evidence on different categories of downturns is necessary. Synthesis of these or similar data in a comparative framework can provide important insights into the causal links between population resilience, risk of exposure and, ultimately, the ability to recover.

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

## Methods

### Calibration and aggregation

Archaeological radiocarbon dates were collated from published studies that previously adopted null hypothesis significance testing (NHST) approaches towards prehistoric demography. Our literature search identified 16 studies that collectively span six continents and approximately 30,000 radiocarbon years (Supplementary Tables 1 and 3–18). We applied a consistent protocol to the calibration of radiocarbon assays. The calibrate function within the R package rcarbon[17] was used to convert [14]C radiocarbon years to calendar years before present. The IntCal20 (ref. 44) and SHCal20 (ref. 45) curves were applied to dates in the northern and southern hemispheres, respectively. Calibrated ages are reported as the 95.4% (two-sigma, $2\sigma$) age range. Laboratory codes and site identification numbers were appended to each calibrated date range and postcalibration distributions were not normalized. To account for between-site variation in sampling intensity, several dates from a single site that are within 50 radiocarbon years of each other were pooled ('binned') before aggregation into regional SPDs.

### Bayesian model fitting

Our protocol aimed to replicate the results of the 16 identified case studies to the greatest extent possible. To reproduce statistically significant negative population events ('downturns') produced by rcarbon's 'modelTest()' function in the original studies, while simultaneously addressing the well-known limitations of using summed probability distributions in NHST, we adopted an alternative, Bayesian modelling approach. Markov Chain Monte Carlo (MCMC) methods implemented in the nimbleCarbon package[46] for radiocarbon data can obtain robust parameter estimates, accounting for radiocarbon measurement errors and sampling error simultaneously. Previously, this has been a major drawback of NHST approaches to aggregated radiocarbon data, with several alternatives proposed in the literature[47–49]. Using the MCMC-derived parameter estimates in posterior predictive checks enabled us to detect periods where expected growth trajectories were lower than the fitted model parameters and which were more robust and accurate than least-squares regression approaches applied directly on SPDs. The protocol produced outputs that are analogous to those in previous NHST studies (Extended Data Fig. 1).

We analysed regional SPDs separately by fitting identical bounded exponential growth models to each dataset. This common model is defined by three parameters: growth rate ($r$) and boundaries ($a$ and $b$). A weakly informative exponentially distributed prior ($\lambda = 1/000.4$) was used for $r$ to capture a broad range of potential growth rates among the cases. Parameters $a$ and $b$ were adopted from the original studies. Markov chain traceplots (Extended Data Figs. 2 and 3) evaluate chain mixing alongside model convergence (Gelman-Rubin $\hat{R}$) and effective sample size diagnostics (Extended Data Table 5). Three chains of 50,000 iterations were run per region, with a burn-in of 5,000 iterations and a thinning interval of 2. To ensure comparability of results with published studies that used a logistic growth model as a null hypothesis, regional datasets were subset based on expert judgement at documented palaeodemographic transitions and treated as two separate exponential growth models. Subsetting was only performed on the Near East and Italy, Sicily and Sardinia datasets. Downturns adjacent to transition points were removed from the sample to avoid including data points introduced by the subsetting. Posterior predictive checks were executed using samples of parameter estimates obtained by the MCMC approach to simulate and back-calibrate a number of radiocarbon dates equal to the sample size. The procedure was repeated 1,000 times to derive critical envelopes.

### Resilience metrics

The resilience metrics target periods when empirical SPD curves are below the 90% confidence envelopes of the fitted models, according to the posterior predictive checks (periods termed 'downturns'). Extraction was performed using a bespoke R function modified from Riris and De Souza[12], which is available at ref. 50. The principal response metrics in our analysis are resistance and resilience (Extended Data Fig. 4). Respectively, these metrics quantify the normalized response to downturns and the rate of recovery relative to baseline conditions. Resistance is measured on SPDs using two parameters: SPD values at the start of a downturn ($b$) and at downturn minima ($x$), whereas resilience is measured across the entire period of decline until its end ($e$) relative to the minimum and baseline (Table 1). Resistance ranges between 0 and 1, indicating a 100% change from baseline to no change. Resilience ranges between −1 and 1, with 1 indicating full recovery by the end of the downturn. Negative values of resilience indicate that the baseline value has been exceeded, although remaining outside the expectations of the null model. Finally, zero indicates no recovery[11,51]. Variations in the shape of the SPD can result under comparable scenarios (that is, with similar demographic dynamics, archaeological sample sizes and disturbances) and hence can produce different results in the metrics because of calibration effects despite their similarity. Although this issue may contribute to noise and error in measurement, it is nevertheless highly unlikely to be systematic or to correlate with other variables of interest. Finally, we conservatively only consider events greater than ten years in duration for our statistical modelling. These events are at an elevated risk of being artefacts of the null model rather than true downturns with an unclear driver.

We also collected information on the start and ends of downturns, their duration, elapsed time until SPD minima were reached, the cumulative number of a downturn and the frequency of downturns (Supplementary Table 2). The frequency of downturns is calculated on a per downturn basis within study regions, using the cumulative number of the downturn, normalized by its duration. We report frequency of downturns for each downturn as a logarithm per millennium.

### Statistical modelling

The target of our comparative analysis is the resistance and resilience of human populations to disturbance as defined in each individual study. Our approach assumed that high values reflect resilient populations that successfully reestablish growth regimes after periods of decline related to disturbance events. We also assumed that downturns are randomly distributed in time and geographical space. To account for the influence of interregional and interevent cultural variation on outcomes, we drew on expert judgement and close readings of the published literature to record the broad category and specific type of disturbance during downturns, as well as the dominant land-use type and the nature of resulting socio-cultural changes, if any (Extended Data Table 1). These variables provided a control on whether a given population within a cultural system retained its identity and function over time, or whether system transformation and adaptive change is archaeologically evident.

Linear mixed-effect models were executed to evaluate the presence and strength of relationships between resistance, resilience, the recorded variables and case study locations. This analysis was performed using the cAIC4 and lme4 R packages[52,53]; scripts are available at ref. 50. Initial models were defined with resistance and resilience as response variables, with only case identifiers (region) as a random effect. As observed downturns were sequential within each case, the random effect controlled for potential pseudoreplication and avoided the need to weight the data by group size. A stepwise search using Akaike's information criterion was implemented for investigating the information gain of including fixed effects in each model in turn. These candidate models were sequentially fitted using restricted maximum likelihood. Most fixed covariates (Extended Data Table 3) were left out of the final models. Region was retained as a random effect in all cases, to produce two models:

$$\text{resistance} \approx (1 \mid \text{region}) + \text{frequency of downturn}$$

$$\text{resilience} \approx (1 \mid \text{region}) + \text{frequency of downturn}$$

Model output is summarized in Extended Data Table 3 and diagnostics are shown in Extended Data Figs. 5 and 6. We present standardized residuals, by region and in full, as well as leverage and Cook's distance.

To further explore the relationship between rates of downturns and resistance and resilience, we performed an additional modelling exercise with the same random effect and full suite of fixed effects, with the frequency of downturn as the independent variable (Extended Data Table 4 and Extended Data Fig. 7).

$$\text{frequency of downturn}$$
$$\approx (1 \mid \text{region}) + \text{land use} + \text{change} + \text{disturbance type} + \text{pace}$$

The effect sizes (standardized coefficients) of the significant model terms are plotted graphically in Fig. 3c and reported in full in Extended Data Table 4. We report effect sizes in the text as $\eta^2$, that is, the total variance explained by differences between means.

### Reporting summary

Further information on research design is available in the Nature Portfolio Reporting Summary linked to this article.

## Data availability

The data supporting the findings of this study are available via Zenodo at https://doi.org/10.5281/zenodo.10061467 (ref. 53).

## Code availability

The code supporting the findings of this study is available via Zenodo at https://doi.org/10.5281/zenodo.10061467 (ref. 53).

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

**Acknowledgements** We thank P. Lockwood for comments on an earlier version of this work. We acknowledge support from the Arts and Humanities Research Council grant AH/X002217/1 (P.R.), Samsung Electronics grant A0342-20220007 (J.B.), Leverhulme Trust grant no. PLP-2019–304 (E.C.) and the Youth Innovation Promotion Association of the Chinese Academy of Sciences grant YIPA-CAS, 2022149 (X.R.).

**Author contributions** P.R. and F.S. conceptualized the study. P.R., E.C., A.P. and F.S. developed the methodology. P.R., B.D., E.K.J., Y.O., A.P., X.R., E.R., P.E.S. and S.S. carried out the investigation. P.R., E.C. and F.S. did the analysis. P.R., J.C.F. and P.E.S. wrote the article. P.R., J.B., E.C., J.C.F., E.K.J., S.Y.M., A.P., E.R., P.E.S., F.S. and S.S. edited the article. P.R., S.Y.M. and E.R. carried out the visualization.

**Competing interests** The authors declare no competing interests.

**Additional information**
**Correspondence and requests for materials** should be addressed to Philip Riris.

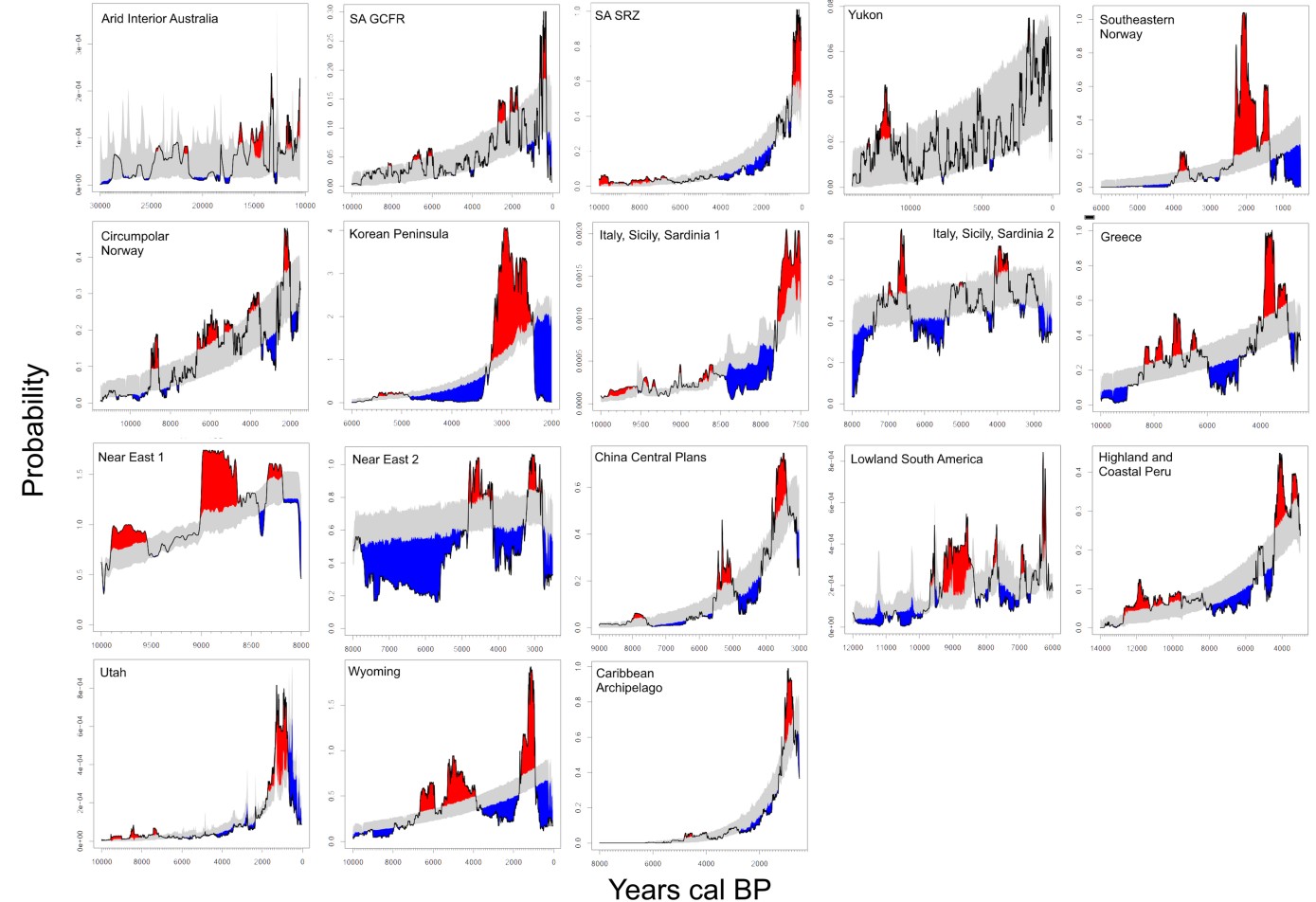

**Extended Data Fig. 1 | Posterior predictive checks for 16 study regions.** Regions shaded in blue indicate period below modelled growth trajectories: downturns.

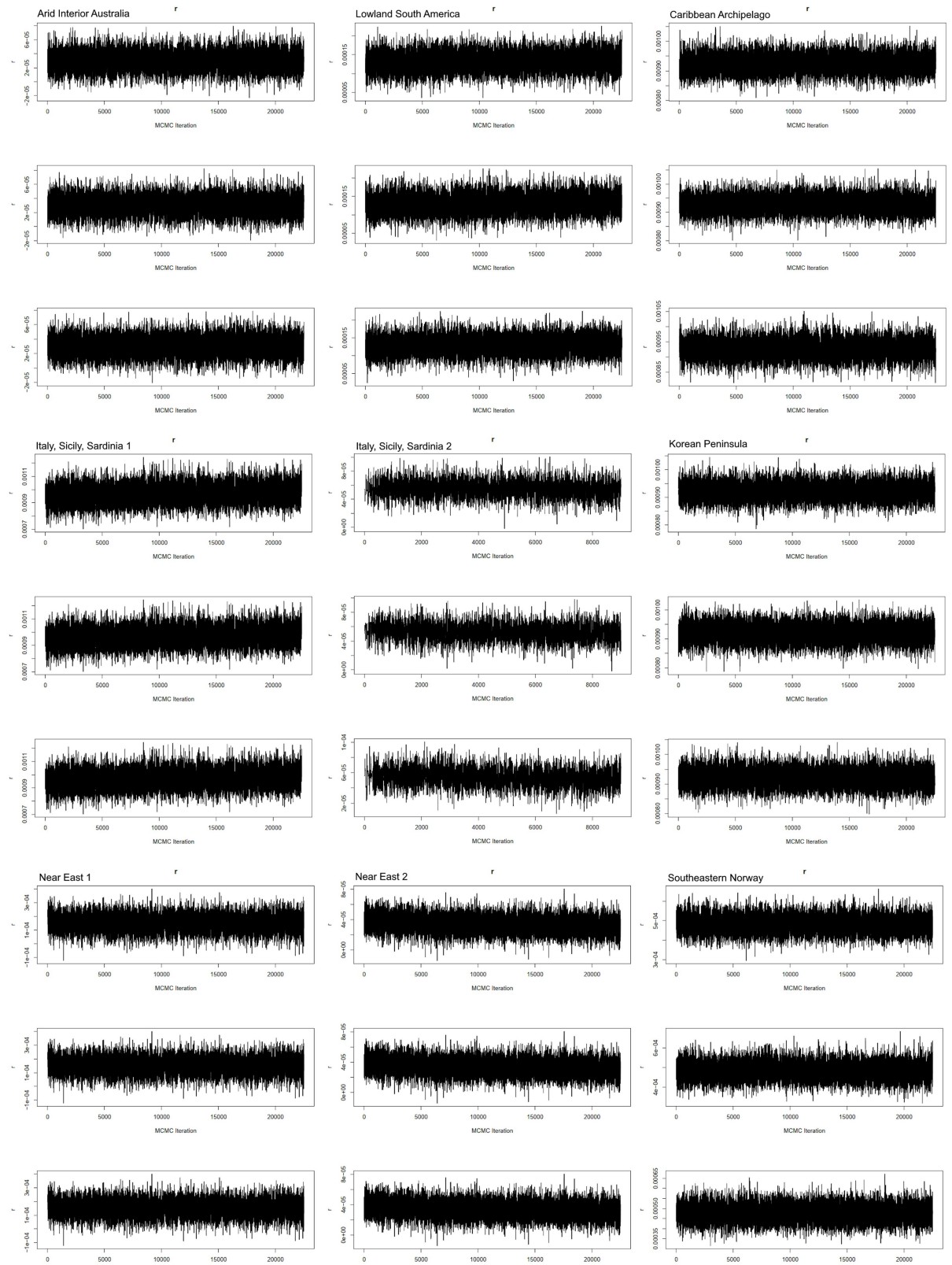

**Extended Data Fig. 2 | Traceplots of chain mixing for the MCMC of each study region and subset.** There is adequate mixing and convergence across chains.

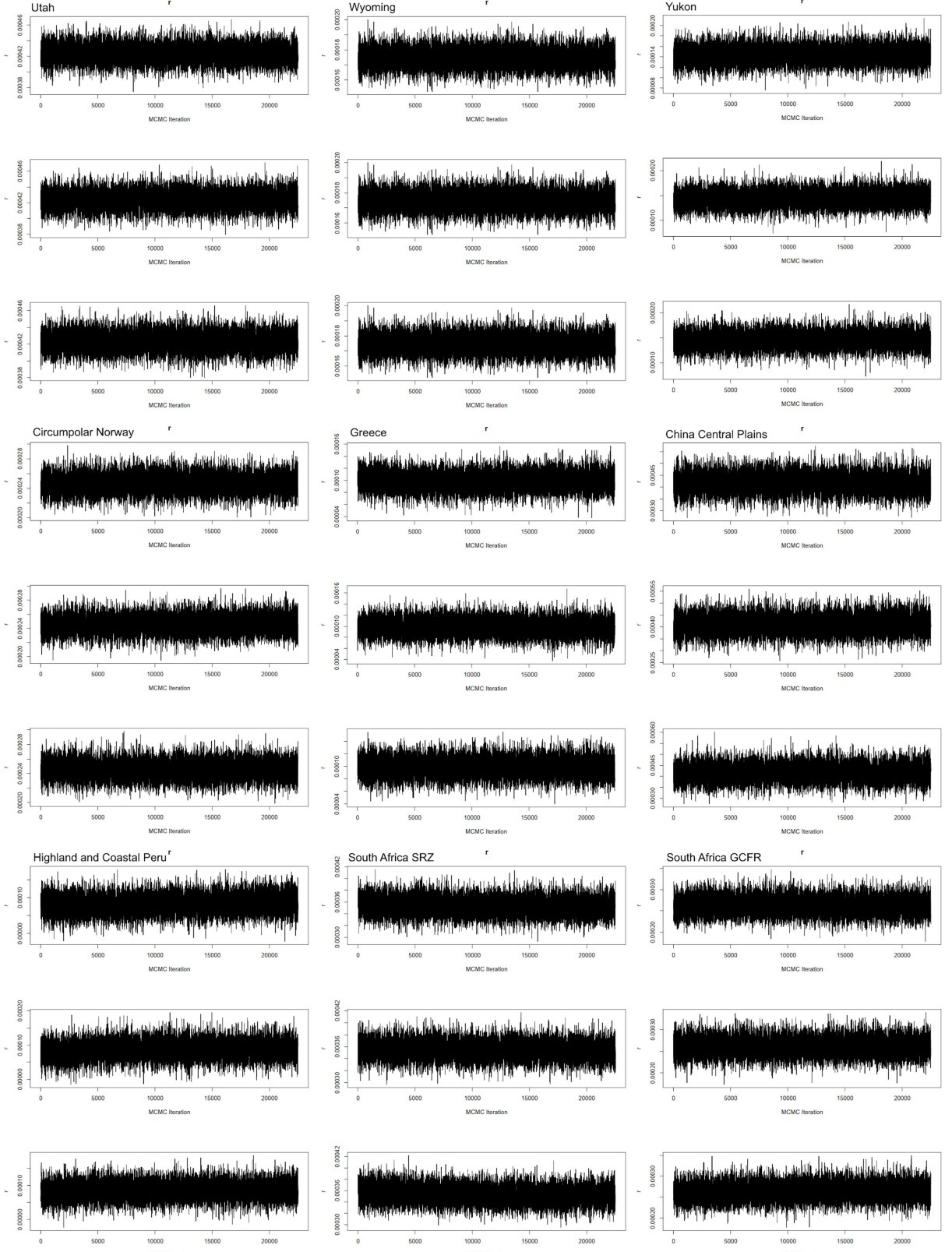

**Extended Data Fig. 3 | Traceplots of chain mixing for the MCMC of each study region and subset.** There is adequate mixing and convergence across chains.

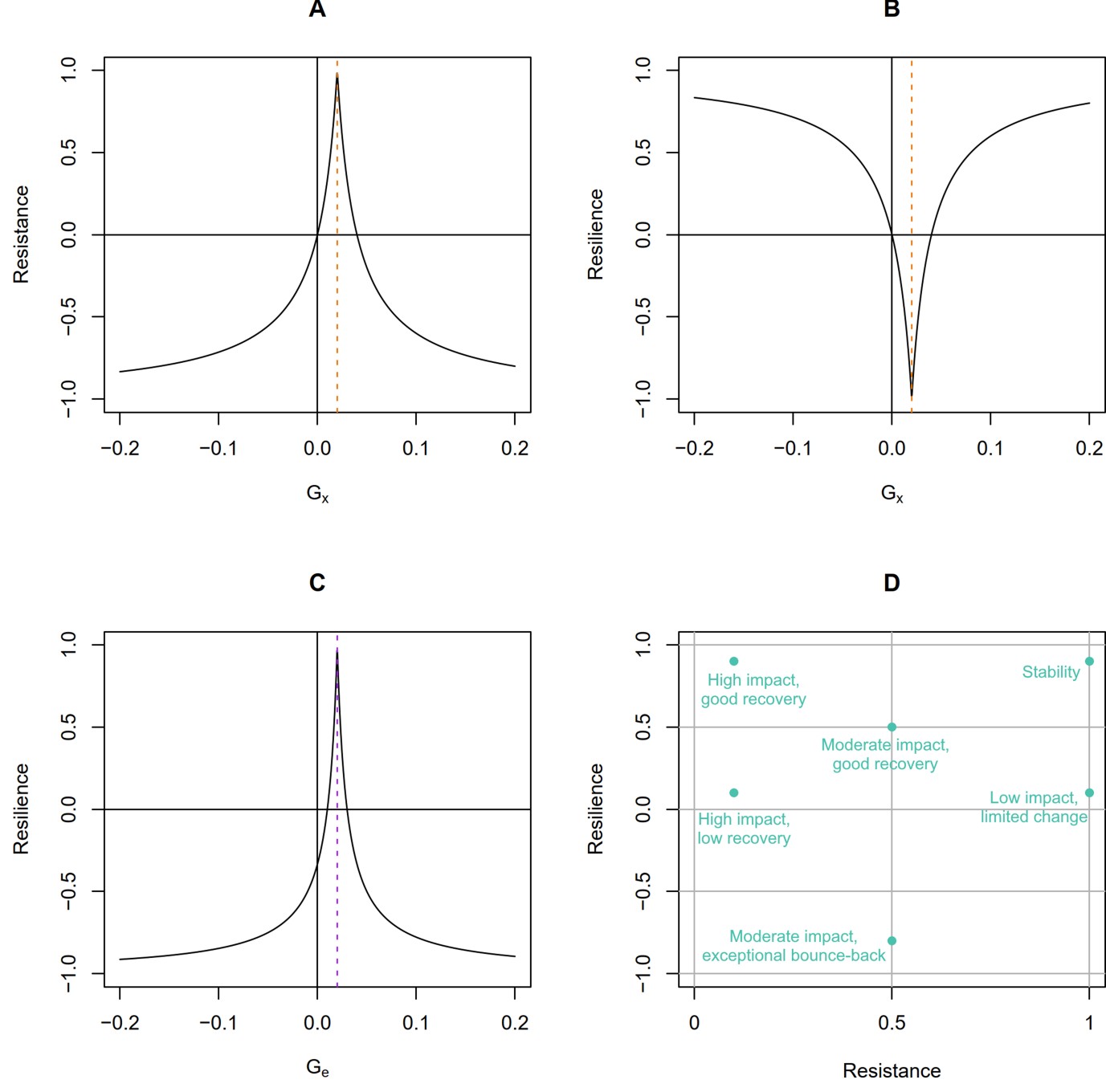

**Extended Data Fig. 4 | Resistance and Resilience as functions of variation in minima ($x$) and end-points ($e$) when baselines ($b$) are held constant (0.02).** (**A**) Resistance equals zero for two different values of $_x$, when $x = 0$ and when $x$ is two times $b$. Negative resistance is theoretically possible but does not occur on SPDs, (**B**) Resilience as a function of $x$, (**C**) Resilience with a varying end point ($e$), and constant start ($b$) and minimum ($x = 0.01$). Resilience equals 1 only when $e = b$, (**D**) Interpretative scatterplot of indicative resistance-resilience outcomes.

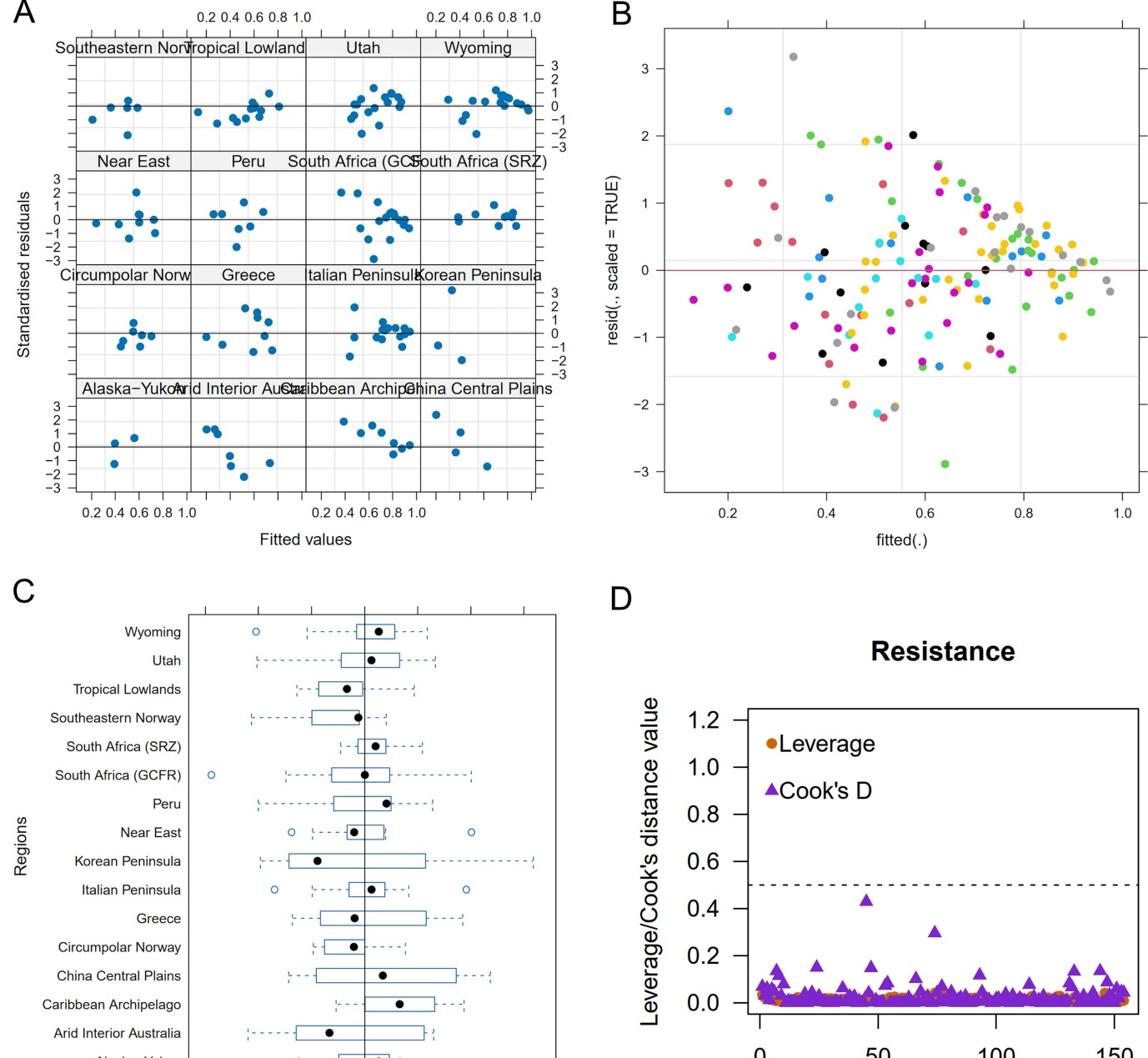

**Extended Data Fig. 5 | Model diagnostics for Model I (Resistance).**
(**A**) Standardised residuals against fitted values for each study region. (**B**) All residuals versus fitted values. (**C**) Standardised residuals by study region for $n = 154$ independent samples across 16 regions. The lower and upper hinges correspond to the 25th and 75th percentiles The upper and lower whiskers extend from the hinges to 1.5 * IQR (where IQR is the inter-quartile range, distance between 25th and 75th percentiles). Data beyond the whiskers are individually plotted outlying points. The black points represent the group median. (**D**) Leverage and Cook's Distance for 154 observations.

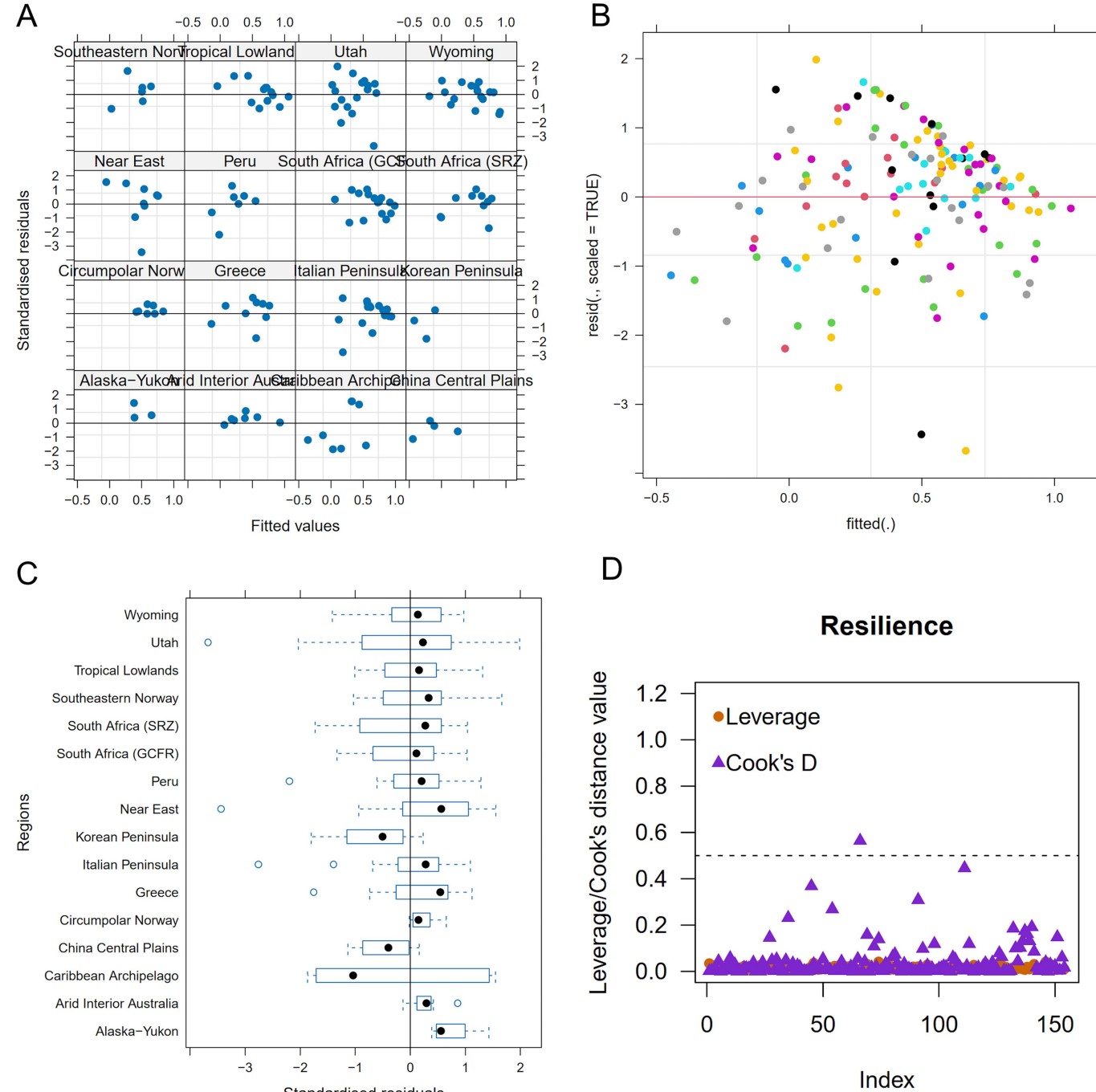

**Extended Data Fig. 6 | Model diagnostics for Model II (Resilience).**
(**A**) Standardised residuals against fitted values for each study region. (**B**) All residuals versus fitted values. (**C**) Standardised residuals by study region for $n = 154$ independent samples across 16 regions. The lower and upper hinges correspond to the 25th and 75th percentiles The upper and lower whiskers extend from the hinges to 1.5 * IQR (where IQR is the inter-quartile range, distance between 25th and 75th percentiles). The black points represent the group median. Data beyond the whiskers are individually plotted outlying points. (**D**) Leverage and Cook's Distance for 154 observations.

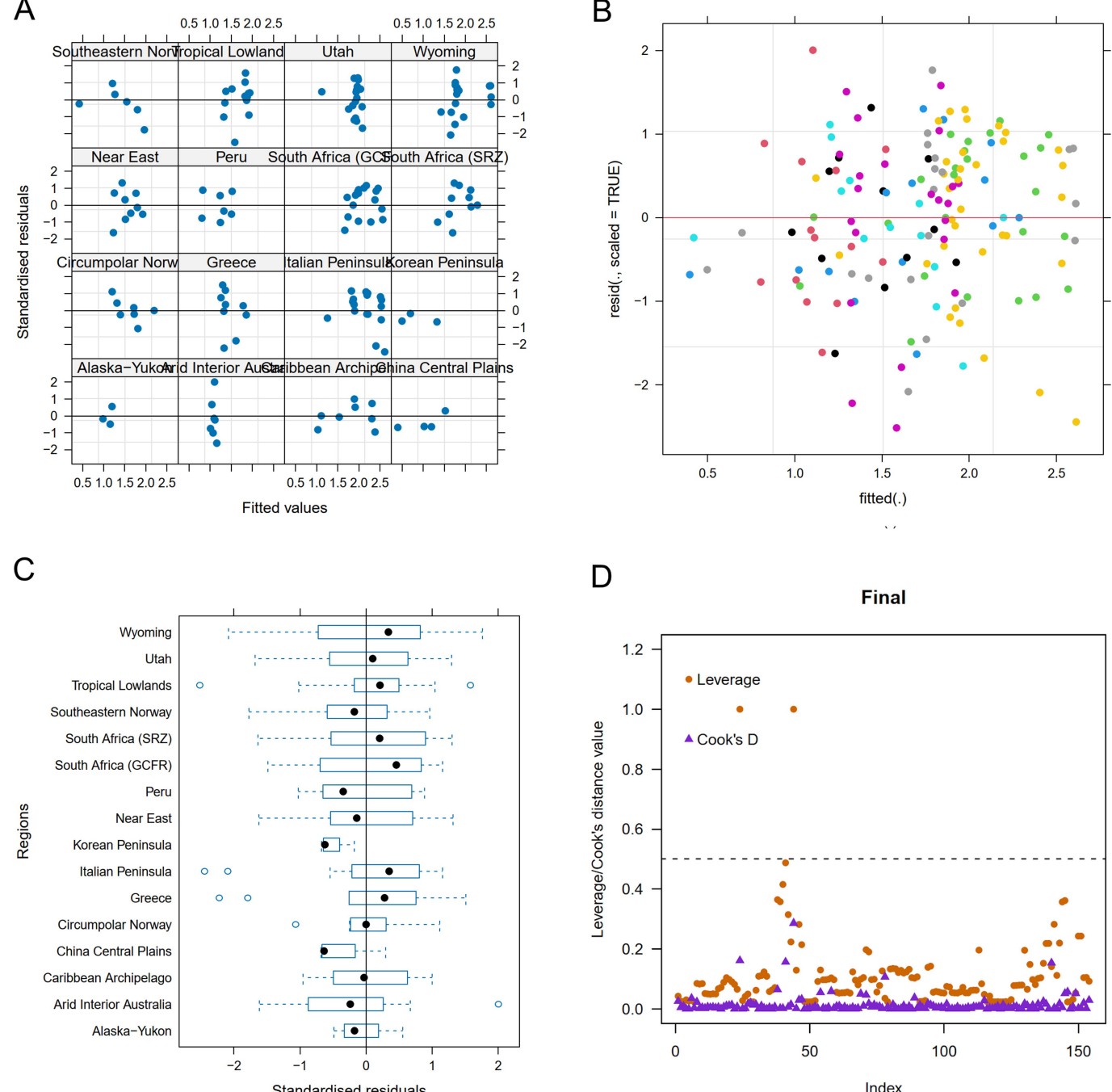

**Extended Data Fig. 7 | Model diagnostics for Model III.** (**A**) Standardised residuals against fitted values for each study region. (**B**) All residuals versus fitted values. (**C**) Standardised residuals by study region for $n = 154$ independent samples across 16 regions. The lower and upper hinges correspond to the 25th and 75th percentiles The upper and lower whiskers extend from the hinges to 1.5*IQR (where IQR is the inter-quartile range, distance between 25th and 75th percentiles). The black points represent the group median. Data beyond the whiskers are individually plotted outlying points. (**D**) Leverage and Cook's Distance for 154 observations.

| Category | Type | Proxies |
|---|---|---|
| Environmental | Aridity | Isotopic geochemistry, sedimentary records, geoarchaeology, climate models |
| | | |
| | Cooling | Sea surface temperature reconstructions, plant micro- and macrofossils, climate models, palaeothermometers |
| | | |
| | Extreme event | Volcanic ash, palaeotsunami records, plant micro- and macrofossils |
| | | |
| | Variability | Hydroclimate records, land use patterns, sedimentary records |
| Cultural | Social reorganisation | Settlement structure and hierarchy, material culture, social networks, technology, warfare |
| | Mobility | Settlement size and duration, relocation frequency, bioarchaeology |
| | | |
| | Colonialism | Archaeological and historical records |
| Mixed | Combinations of the above environmental and cultural factors | Intersecting of compounding vulnerabilities |
| | | |
| | Carrying capacity | Settlement consolidation/abandonment, subsistence change to tolerant crops, 'boom-and-bust', shifts in resource diversity |
| Unclear | | |
| | | Downturns with lack of evidence for a clear driver |

**Extended Data Table 2 | Initial mixed-effect model diagnostics for Resistance and Resilience with Study Region included as a random effect**

| Predictors | Resistance | | |
|---|---|---|---|
| | Estimates | CI | p |
| (Intercept) | 0.43 | 0.14 – 0.71 | 0.003 |
| Name [Arid Interior Australia] | -0.08 | -0.42 – 0.26 | 0.637 |
| Name [Caribbean Archipelago] | 0.42 | 0.09 – 0.76 | 0.013 |
| Name [China Central Plains] | 0.06 | -0.32 – 0.43 | 0.759 |
| Name [Circumpolar Norway] | 0.08 | -0.26 – 0.42 | 0.644 |
| Name [Greece] | 0.17 | -0.16 – 0.50 | 0.302 |
| Name [Italian Peninsula] | 0.33 | 0.02 – 0.64 | 0.036 |
| Name [Korean Peninsula] | -0.08 | -0.48 – 0.32 | 0.684 |
| Name [Near East] | 0.12 | -0.21 – 0.45 | 0.464 |
| Name [Peru] | 0.03 | -0.31 – 0.36 | 0.877 |
| Name [South Africa (GCFR)] | 0.29 | -0.02 – 0.60 | 0.063 |
| Name [South Africa (SRZ)] | 0.3 | -0.03 – 0.62 | 0.072 |
| Name [Southeastern Norway] | -0.09 | -0.44 – 0.26 | 0.611 |
| Name [Tropical Lowlands] | 0.04 | -0.28 – 0.35 | 0.823 |
| Name [Utah] | 0.23 | -0.07 – 0.54 | 0.135 |
| Name [Wyoming] | 0.29 | -0.01 – 0.60 | 0.062 |

| Random Effects | |
|---|---|
| $\sigma^2$ | 0.06 |
| $\tau_{00}$ Name | 0 |
| N Name | 16 |
| Observations | 154 |
| Marginal R2 / Conditional R2 | 0.258 / NA |

| Predictors | Resilience | | |
|---|---|---|---|
| | Estimates | CI | p |
| (Intercept) | 0.81 | 0.27 – 1.35 | 0.004 |
| Name [Arid Interior Australia] | -0.3 | -0.94 – 0.35 | 0.367 |
| Name [Caribbean Archipelago] | -0.8 | -1.43 – -0.16 | 0.014 |
| Name [China Central Plains] | -1.12 | -1.84 – -0.40 | 0.002 |
| Name [Circumpolar Norway] | -0.11 | -0.75 – 0.54 | 0.748 |
| Name [Greece] | -0.31 | -0.94 – 0.32 | 0.329 |
| Name [Italian Peninsula] | -0.2 | -0.79 – 0.39 | 0.502 |
| Name [Korean Peninsula] | -1.36 | -2.12 – -0.59 | 0.001 |
| Name [Near East] | -0.31 | -0.93 – 0.32 | 0.334 |
| Name [Peru] | -0.62 | -1.27 – 0.03 | 0.061 |
| Name [South Africa (GCFR)] | -0.17 | -0.76 – 0.42 | 0.566 |
| Name [South Africa (SRZ)] | -0.36 | -0.98 – 0.25 | 0.247 |
| Name [Southeastern Norway] | -0.3 | -0.96 – 0.36 | 0.374 |
| Name [Tropical Lowlands] | -0.14 | -0.74 – 0.46 | 0.653 |
| Name [Utah] | -0.49 | -1.08 – 0.10 | 0.102 |
| Name [Wyoming] | -0.38 | -0.96 – 0.21 | 0.208 |

| Random Effects | |
|---|---|
| $\sigma^2$ | 0.23 |
| $\tau_{00}$ Name | 0 |
| N Name | 16 |
| Observations | 154 |
| Marginal R2 / Conditional R2 | 0.229 / NA |

Significant terms (p<0.05) based on a two-sided test in **bold**.

**Extended Data Table 3 | Final mixed-effect model diagnostics for Resistance and Resilience with Study Region included as a random effect and Frequency of downturn as a fixed effect**

| Predictors | Resistance | | |
|---|---|---|---|
| | Estimates | CI | p |
| (Intercept) | 0.18 | 0.09 – 0.27 | <0.001 |
| Rate of downturn | 0.25 | 0.20 – 0.30 | <0.001 |
| Random Effects | | | |
| σ2 | 0.04 | | |
| τ00 Name | 0 | | |
| ICC | 0.06 | | |
| N Name | 16 | | |
| Observations | 154 | | |
| Marginal R2 / Conditional R2 | 0.431 / 0.466 | | |

| Predictors | Resilience | | |
|---|---|---|---|
| | Estimates | CI | p |
| (Intercept) | -0.24 | -0.45 – -0.03 | 0.025 |
| Rate of downturn | 0.41 | 0.30 – 0.51 | <0.001 |
| Random Effects | | | |
| σ2 | 0.18 | | |
| τ00 Name | 0.05 | | |
| ICC | 0.21 | | |
| N Name | 16 | | |
| Observations | 154 | | |
| Marginal R2 / Conditional R2 | 0.291 / 0.441 | | |

Significant terms (p < 0.05) based on a two-sided test in **bold**.

**Extended Data Table 4 | Mixed-effect diagnostics for Rate of downturns with Study Region included as a random effect and Land Use, Change, Disturbance Type, and Pace as a fixed effects**

| Predictors | Estimates | CI | p |
|---|---|---|---|
| (Intercept) | 1.02 | 0.45 – 1.59 | 0.001 |
| Landuse deriv [Agriculture] | 0.82 | 0.37 – 1.26 | <0.001 |
| Landuse deriv [Agropastoralism] | 0.48 | 0.02 – 0.94 | 0.043 |
| Landuse deriv [Low-level food production] | 0.18 | -0.36 – 0.72 | 0.511 |
| Landuse deriv [Maritime forager] | 0.46 | -0.53 – 1.45 | 0.356 |
| Landuse deriv [Mixed] | 0.2 | -0.19 – 0.59 | 0.31 |
| Change deriv [1] | -0.28 | -0.60 – 0.05 | 0.094 |
| Type deriv [Carrying capacity] | -0.01 | -0.61 – 0.59 | 0.966 |
| Type deriv [Colonialism] | -0.47 | -1.79 – 0.85 | 0.484 |
| Type deriv [Cooling] | -0.13 | -0.92 – 0.65 | 0.734 |
| Type deriv [Extreme event] | 0.1 | -0.77 – 0.97 | 0.817 |
| Type deriv [Mobility] | 1.05 | 0.45 – 1.65 | 0.001 |
| Type deriv [Overshoot] | 0.81 | -0.59 – 2.22 | 0.255 |
| Type deriv [Reorganisation] | 0.56 | -0.09 – 1.21 | 0.093 |
| Type deriv [U] | 0.24 | -0.14 – 0.63 | 0.217 |
| Type deriv [Variability] | 0.7 | 0.23 – 1.17 | 0.004 |
| LD | 0.27 | -0.16 – 0.70 | 0.212 |
| Random Effects | | | |
| σ2 | 0.36 | | |
| τ00 Name | 0.2 | | |
| ICC | 0.35 | | |
| N Name | 16 | | |
| Observations | 154 | | |
| Marginal R2 / Conditional R2 | 0.188 / 0.475 | | |

Significant terms (p<0.05) based on a two-sided test in **bold**.

**Extended Data Table 5 | Convergence (Gelman-Rubin Ȓ) and Effective Sample Size diagnostics for MCMC fits**

| Name | Rhat | Upper CI | Effective sample size |
|---|---|---|---|
| Circumpolar Norway | 1.002623 | 1.009287 | 29380 |
| Arid Interior Australia | 1.000626 | 1.00223 | 29500 |
| Greece | 1.004162 | 1.01486 | 29333 |
| China Central Plains | 1.003165 | 1.011655 | 27797 |
| Italy, Sicily, and Sardinia 1 | 1.00056 | 1.00168 | 28913 |
| Italy, Sicily, and Sardinia 2 | 1.008331 | 1.029792 | 10834 |
| Korean Peninsula | 1.002227 | 1.008151 | 29757 |
| Lowland South America | 1.001278 | 1.004562 | 28951 |
| Near East 1 | 1.005172 | 1.011312 | 28652 |
| Near East 2 | 1.008798 | 1.019199 | 26412 |
| Highland and Coastal Peru | 1.00248 | 1.008801 | 22778 |
| Southern Africa Greater Cape Floristic Region | 1.008906 | 1.032177 | 29397 |
| Southern Africa Summer Rainfall Zone | 1.006787 | 1.024687 | 25185 |
| Southeastern Norway | 1.003775 | 1.01364 | 28322 |
| Utah | 1.000845 | 1.00316 | 29660 |
| Wyoming | 1.00006 | 1.00018 | 28909 |
| Yukon | 1.001744 | 1.00636 | 28998 |
| Caribbean Archipelago | 1.003283 | 1.011956 | 28003 |

The Gelman-Rubin convergence diagnostic provides a convergence summary based on multiple chains, in our case, three per MCMC fit.

# Reporting Summary

## Statistics

For all statistical analyses, confirm that the following items are present in the figure legend, table legend, main text, or Methods section.

| n/a | Confirmed | |
|---|---|---|
| ☐ | ☒ | The exact sample size (*n*) for each experimental group/condition, given as a discrete number and unit of measurement |
| ☐ | ☒ | A statement on whether measurements were taken from distinct samples or whether the same sample was measured repeatedly |
| ☐ | ☒ | The statistical test(s) used AND whether they are one- or two-sided *Only common tests should be described solely by name; describe more complex techniques in the Methods section.* |
| ☐ | ☒ | A description of all covariates tested |
| ☐ | ☒ | A description of any assumptions or corrections, such as tests of normality and adjustment for multiple comparisons |
| ☐ | ☒ | A full description of the statistical parameters including central tendency (e.g. means) or other basic estimates (e.g. regression coefficient) AND variation (e.g. standard deviation) or associated estimates of uncertainty (e.g. confidence intervals) |
| ☐ | ☒ | For null hypothesis testing, the test statistic (e.g. *F*, *t*, *r*) with confidence intervals, effect sizes, degrees of freedom and *P* value noted *Give P values as exact values whenever suitable.* |
| ☐ | ☒ | For Bayesian analysis, information on the choice of priors and Markov chain Monte Carlo settings |
| ☐ | ☒ | For hierarchical and complex designs, identification of the appropriate level for tests and full reporting of outcomes |
| ☒ | ☐ | Estimates of effect sizes (e.g. Cohen's *d*, Pearson's *r*), indicating how they were calculated |

*Our web collection on statistics for biologists contains articles on many of the points above.*

## Software and code

Policy information about availability of computer code

| | |
|---|---|
| Data collection | No code was used. |
| Data analysis | The code supporting the findings of this study are available in Zenodo with the identifier doi: 10.5281/zenodo.10061467 (https://dx.doi.org/10.5281/zenodo.10061467). |

For manuscripts utilizing custom algorithms or software that are central to the research but not yet described in published literature, software must be made available to editors and reviewers. We strongly encourage code deposition in a community repository (e.g. GitHub). See the Nature Portfolio guidelines for submitting code & software for further information.

## Data

Policy information about availability of data

All manuscripts must include a data availability statement. This statement should provide the following information, where applicable:

- Accession codes, unique identifiers, or web links for publicly available datasets
- A description of any restrictions on data availability
- For clinical datasets or third party data, please ensure that the statement adheres to our policy

The data supporting the findings of this study are available in Zenodo with the identifier doi: 10.5281/zenodo.10061467 (https://dx.doi.org/10.5281/zenodo.10061467).

# Research involving human participants, their data, or biological material

Policy information about studies with <u>human participants or human data</u>. See also policy information about <u>sex, gender (identity/presentation), and sexual orientation</u> and <u>race, ethnicity and racism</u>.

| | |
|---|---|
| Reporting on sex and gender | NA |
| Reporting on race, ethnicity, or other socially relevant groupings | NA |
| Population characteristics | NA |
| Recruitment | NA |
| Ethics oversight | NA |

Note that full information on the approval of the study protocol must also be provided in the manuscript.

# Field-specific reporting

Please select the one below that is the best fit for your research. If you are not sure, read the appropriate sections before making your selection.

☐ Life sciences ☒ Behavioural & social sciences ☐ Ecological, evolutionary & environmental sciences

For a reference copy of the document with all sections, see <u>nature.com/documents/nr-reporting-summary-flat.pdf</u>

# Behavioural & social sciences study design

All studies must disclose on these points even when the disclosure is negative.

| | |
|---|---|
| Study description | Quantitative meta-analysis of archaeological radiocarbon data. |
| Research sample | Approx. 40,000 georeferenced radiocarbon dates over 16 global study locations collected from the extant research literature, which are representative of the spatio-temporal variability in land use, environment, social organisation, technology, political complexity, and extent of prehistoric human populations. In each study region, the number of radiocarbon dates are an accurate reflection of the total number of dates made on archaeological remains. |
| Sampling strategy | Studies were reviewed based on three criteria: evidence for significant downturns, their scope, and the inclusion of radiocarbon datasets. A lack of any single criterion resulted in the exclusion of a study. Cases with no reported downturns were not included, nor were those whose scope was restricted to specific activities within a regional radiocarbon dataset, such as flint mining. No sample size calculation was performed, however, our smallest regional sample size (n=272) is sufficient to replicate original study results and extract the features of population variability over time that are of interest to the study. All radiocarbon samples were included, where they reflect an archaeologically dated depositional event. |
| Data collection | We included all radiocarbon data reported by the authors of the original study. Where published data have been superseded by later compilations, dates were added from the People3k database, a systematic compilation of cleaned radiocarbon dates, based on the geographical area of the original study, and duplicates (based on laboratory code) were removed. Data were collected and collated with Microsoft Excel. The researcher was not blinded to experimental condition or the study hypothesis, insofar as this is applicable. |
| Timing | September 2021 - January 2022. |
| Data exclusions | No data were excluded. |
| Non-participation | Not applicable. |
| Randomization | Not applicable. |

# Reporting for specific materials, systems and methods

We require information from authors about some types of materials, experimental systems and methods used in many studies. Here, indicate whether each material, system or method listed is relevant to your study. If you are not sure if a list item applies to your research, read the appropriate section before selecting a response.

## Materials & experimental systems

| n/a | Involved in the study |
|-----|----------------------|
| ☒ | Antibodies |
| ☒ | Eukaryotic cell lines |
| ☒ | Palaeontology and archaeology |
| ☒ | Animals and other organisms |
| ☒ | Clinical data |
| ☒ | Dual use research of concern |
| ☒ | Plants |

## Methods

| n/a | Involved in the study |
|-----|----------------------|
| ☒ | ChIP-seq |
| ☒ | Flow cytometry |
| ☒ | MRI-based neuroimaging |

## Plants

| | |
|---|---|
| Seed stocks | NA |
| Novel plant genotypes | NA |
| Authentication | NA |

