## [Peer Review File · Nature]

Manuscript Title: Frequent disturbances enhanced the resilience of past human populations

Reviewer Comments & Author Rebuttals

Reviewer Reports on the Initial Version:

Referees' comments:

Referee #1 (Remarks to the Author):

The authors use population dynamics reconstructions (based on summed probability distributions of radiocarbon dates) from 16 regions to explore which factors influence the resilience and resistance of human communities worldwide and with deep temporal scope. The main conclusion is that the frequency of population downturn events significantly impacts the resilience and resistance of human populations. The results also reveal the average durations of downturn events, as well as the distribution of resilience and resistance values. The authors also show that agriculture is associated with more frequent downturn events, which in turn increase resilience of human populations in the long run.

This research utilizes the wealth of archaeological data rigorously and systematically, with implications of direct relevance to the issues of global significance in the present, such as resilience and resistance in the face of environmental and cultural change. The statistical approach is state of the art, and the results of the data analysis support the conclusions. It is an impressive and exemplary study both in conception and execution, so I strongly recommend publishing this paper.

I see no significant flaws in the research design and execution, but the paper would benefit from clarifying and addressing minor technical issues:

1) The resilience/resistance metrics are based on the SPD curve, and the shape of the SPD curve depends on the underlying demographic dynamics and the shape of the calibration curve. The latter is going to distort the actual shape of the population curve. For example, suppose we have two identical scenarios of downturns in different periods, and we have the same number of radiocarbon dates for each scenario. In that case, the SPD curves will look different even though the underlying demographic dynamics are the same, i.e., with the same resistance and resilience values. This distortion is not guaranteed to be linear, so relying on the ratios of SPD values will not solve the problem. I do not think that this is a significant issue, as this effect is highly unlikely to be systematic or to correlate with other variables of interest. However, it contributes to the noise and error in measuring resilience, resistance, and related variables, so it should be acknowledged as such.

2) Could the authors, please, clarify why resistance and resilience are calculated in the way they are. It seems to me that the most intuitive way to measure resistance in a relative way is to calculate the ratio of the difference between G_b and G_x and the value of the G_b . Likewise, it is more intuitive to look at the G_e/G_b ratio for resilience. What is the logic behind the formulae in Table 1?

3) Based on Figure 1B, G_b value is chosen to be the value on the bottom of the 95% SPD null model envelope rather than the peak SPD value preceding the downturn, and G_e value is chosen similarly, not as a peak after the downturn, but as the value on the curve where the curve intersects with the 95% envelope. Does this affect the conclusion that in most cases the baseline population size is not attained after the downturn?

4) Lines 103-104: The authors demonstrate that mobility is the second most common type of disturbance. Mobility can influence changes in the SPD curve in the absence of changes in

population size. If the population switches from a more mobile to a less mobile way of living, this will be reflected as a downturn in the SPD curve. Is this kind of effect taken into consideration?

5) If I understood correctly, for each downturn event, resilience and resistance are measured based on the SPD curve section associated with the event, and the cumulative frequency of downturns up to a particular downturn is associated with the focal event. If this is correct (it seems to make most sense and this is what entries in Table S2 suggest), this should be more clearly stated, as it is easy to misunderstand that the FD is calculated at the regional level which would result in only 16 possible FD values.

6) Line 378 and line 384. A figure number is missing.

7) Identification of downturns. I saw in the supplementary files that the `resmet()` function is mentioned which performs such tests on the SPD curve to identify significant downturns. I did not quite understand how is this test different from the Edinborough et al. (2017) test? All this should be clarified as the number of downturn events depends on the method of their identification. Moreover, downturns can happen in time intervals when the entire SPD curve is in the "red zone", above the upper 95% envelope limit. Why are these not taken into account?

8) The main text states that the total number of downturns is 154, yet in Table S2 and the supplementary CSV file, there are 169 downturns. What is the reason for this mismatch?

9) I had some troubles with running the code for the Norway example. When I ran the lines from the "Run model" section, the following error message appeared:
Error in `checkForRemoteErrors(val)` :
one node produced an error: Failed to create the shared library. Run '`printErrors()`' to see the compilation errors.

Yet when I ran the `printErrors` command, the message was that there were no errors. But I could not generate the output.

Referee #2 (Remarks to the Author):

This article presents the results of a synthetic analysis of data drawn from a range of socio-cultural systems for which archaeological and palaeoecological data exists and covering some 30,000 years of prehistory. The aim is to determine the degree to which evidence for resistance to and recovery from disturbance (as qualified in Fig 1c) indicates common patterns and if so how these are manifested in the archaeological record. Such evidence indicates fluctuations in population or archaeological activity in prehistoric systems, as reflected in calibrated radiocarbon dates over the long term (centennial and higher). The analysis suggests that the more often a population has to deal with disturbances the greater its capacity to resist and recover from later downturns, although also noting that resistance and recovery are not necessarily commensurate. One outcome of the analysis is to note that agro-pastoral societies, while subject to a higher incidence of disturbance, also indicate (in general) greater levels of resilience/recovery.

This is an original and innovative study which champions an approach that would question the general tendency to particularize socio-ecological systems and make comparisons at the short- and medium-term level rather than seek common systemic elements over the much longer term. The statistical analysis of the data and the correlations drawn from it regarding disturbance, resistance and resilience are impressive and provide an important basis for developing and expanding the discussion, and the Bayesian approach contributes considerably to avoiding simplistic or reductionist statistical correlations. Focusing on the very long term has the advantage of revealing broader and more generalizable systemic patterns which cannot be

detected through shorter-term more particularistic analyses of recent social-cultural and socio-ecological systems.

Within the framework set by the authors in the opening sections the argument seems to me relatively robust. In the Discussion the authors translate the statistical correlations into a few illustrative historical examples and note that the result of their analysis "runs counter to the historical particularism of archaeological work on past resilience, which often emphasises the contingencies, decisions, and practices that underwrote successful adaptations in specific times and places." I am not persuaded that this is necessarily the case - contingency and the resultant emergent practices can be taken within a broader framework of causal relationships that still allows for specificity in time and place, and those specificities are essential to more detailed comparative analysis. Identifying the common mechanism is therefore the challenge, as they correctly state - but if it is too broad or general - i.e. simply a phylogenetically behavioural feature of homo sapiens then it doesn't really help explain the specificity of outcomes of historical cases of resilience and recovery, nor does it do more than describe the 'human condition'. So it would be helpful to understand more about how the implications of such a very-long-term analysis feed into thinking about contemporary policy and future planning scenarios. This is not a criticism of the paper, however, merely something that could be developed further in due course. One might argue, e.g., that since the types of SESs that experience more frequent downturns are largely agro-pastoral, and hence likely more complexly articulated, the rate at which they exploit their environments and potentially contribute to imbalances that themselves lead to social-economic and social-ecological tensions is going to be higher than for, e.g. hunter-gather groups, i.e. 'simpler' or less hierarchically complex societies. By the same token, higher levels of socio-economic differentiation and exploitation may facilitate more effective managerial responses over the medium term, thus permitting faster recovery. So the general mechanism may well be tied in with degrees of complexity or modes of articulation of elements (ideological, technical, organizational) within complex societal systems. Finally, although again not a criticism of this paper, in terms of important insights into the causal links between population resilience, risk of exposure, and ultimately, the ability to recover, the social costs of resilience at this level of generalisable comparative analysis remain invisible, yet these must surely be seen as crucial for any actionable policy, and need to be written into the model through culture-specific analysis of the data on the ground if they are to have any longer-term purchase on policy-making.

In terms of references, lucidity and clarity of argument, coherence of the discussion as well as appropriateness of abstract, introduction and conclusion - all fine.

In short - a stimulating, innovative and challenging article that thoroughly deserves publication.

John Haldon

Author Rebuttals to Initial Comments:

We find the comments of the reviewers universally helpful in improving the manuscript and thank them for their consideration. Reviewer 1 highlights some salient points in their review, our responses to these are given in blue Times New Roman font.

In the case of Reviewer 2, we have amended the manuscript Discussion to accommodate their comments on historical particularism. Regarding their observations that the rate of environmental exploitation of agricultural/agropastoral societies likely contributed to a more rapid propagation of socio-ecological imbalances, we fully acknowledge that such a mechanism may be underlying the results we have presented. Similarly, we agree with the suggestion that the degree of complexity/integration of a given population is also likely to be crucial in determining rates of recovery over the medium term. We have attempted to incorporate these perspectives, however, we feel that we presently lack the data that would justify stronger conclusions on these two themes. They are nevertheless plausible scenarios from which testable hypotheses can be derived, and we hope to extend future analyses to the pursuit of these questions.

Overall, we appreciate the profundity of this reviewer's insights on the subject of policy impact, which we anticipate will strongly inform future pursuits in this vein.

Reviewer 1

1) The resilience/resistance metrics are based on the SPD curve, and the shape of the SPD curve depends on the underlying demographic dynamics and the shape of the calibration curve. The latter is going to distort the actual shape of the population curve. For example, suppose we have two identical scenarios of downturns in different periods, and we have the same number of radiocarbon dates for each scenario. In that case, the SPD curves will look different even though the underlying demographic dynamics are the same, i.e., with the same resistance and resilience values. This distortion is not guaranteed to be linear, so relying on the ratios of SPD values will not solve the problem. I do not think that this is a significant issue, as this effect is highly unlikely to be systematic or to correlate with other variables of interest. However, it contributes to the noise and error in measuring resilience, resistance, and related variables, so it should be acknowledged as such.

We have amended the text to acknowledge there may be some noise in the determination of these metrics. We would like to note that the publication that introduced these metrics to SPDs (Riris and De Souza 2021, *Frontiers*) directly investigated the impact of varying the start/end-points of downturns. Although some variability is evident (their Figure 5), particularly with Resilience due to its comparison of three parameters to Resistance's two, the downturns they detect on different populations experiencing the same disturbance nonetheless appear as fully distinct. We therefore agree that such noise is unlikely to systematically bias the results.

2) Could the authors, please, clarify why resistance and resilience are calculated in the way they are. It seems to me that the most intuitive way to measure resistance in a relative way is to calculate the ratio of the difference between G_b and G_x and the value of the G_b . Likewise, it is more intuitive to look at the G_e/G_b ratio for resilience. What is the logic behind the formulae in Table 1?

The aim of the metrics is to enable comparisons between downturns. The formulae themselves are constructed so that:

- 1) the index is bound, as stated in the Methods, between -1 and 1. In practical terms, on SPDs, Resistance is limited to 0 and 1 as negative values are not possible in probability distributions.
- 2) Additionally, the indices cannot tend to infinity, due to the way the denominator is constructed, as it cannot result in a divide by zero error as raw SPD values could, as proposed by the reviewer.
- 3) Both metrics are standardised by the pre-downturn conditions of the population proxy, while resilience is additionally standardised by the amount of change to the proxy experienced during the downturn.

Figure S4 illustrates these advantageous qualities. We have taken the reviewer’s suggestion and produced analogous plots using their definitions, below:

Figure 1: Results of suggested definitions of Resistance and Resilience. Plots display metrics as functions of variation in minima (x) and end-points (e) while baselines (b) are held constant (0.02). A) Resistance is not bounded but varies monotonically. B & C) Resilience is not bounded and can tend towards infinity.

We show the advantage of adopting the indices employed in the manuscript (following Riris and De Souza 2021) over the reviewer’s suggestions with a simple comparative illustration. Here, we have taken two downturns from our dataset (South Africa GCFR, UniqID 200002 & Peru, UniqID 1300004) with very similar Resistance and Resilience values per our formulae, as well as similar durations. The downturns bear a qualitative similarity in the SPD shape additionally.

Table 1 illustrates the metrics we derived, in comparison to the suggested metrics. The suggested metrics return a higher Resilience for the GCFR in comparison to our metrics, in which Peru has a value 66% higher than the GCFR. Resistance, though higher in both suggested and actual metrics for the GCFR, is an order of magnitude higher than Peru in the suggested metrics, whereas our metrics are comparable.

Table 1: Suggested and reported metrics for Resistance and Resilience on two similar downturns.

UniqID	b	x	e	Resistance	Resilience	Suggested resistance	Suggested resilience
200002 GCFR	0.00002565	0.00002295	0.00004141	0.788853	0.196325	0.105	1.614
1300004	0.00006355	0.00006148	0.00006646	0.783256	0.285709	0.033	1.046

Figure 2: Downturns 130004 (Peru, top) and 200002 (GCFR, bottom). These are also visible in Figure S1 of the manuscript.

Analytically, we believe that our metrics achieve the outcome that the reviewer correctly identifies as desirable: resistance indicates the deviation of the proxy from baseline, standardised relative to the baseline, while resilience does the same while also standardised by the total relative change. We agree that the reviewer’s suggestions follow intuition, however, practical considerations in the handling of numerical data must also be taken into account as illustrated in Figure 1. The cases shown in this figure (evidencing a lack of bounding, tendencies towards infinity, lack of standardisation) illustrate how the suggested metrics would undermine the comparability of our results.

Our approach provides explicitly defined, quantitative, and reproducible methods, which accompany this paper for future use and refinement. Although any metric or point estimate will imperfectly reflect a complex reality, the tests we have adapted and implemented have a pedigree in quantitative ecology, with the analytical benefits we have espoused in our manuscript (Nimmo et al. 2015; Van Meerbeek et al. 2021). Our approach enables the archaeological study of resilience to locate commonalities of language with the ecological resilience literature. The code for reproducing the above can be found at the following URL: <https://pastebin.com/hxYAEA5D>

3) Based on Figure 1B, Gb value is chosen to be the value on the bottom of the 95% SPD null model envelope rather than the peak SPD value preceding the downturn, and Ge value is chosen similarly, not as a peak after the downturn, but as the value on the curve where the curve intersects with the 95% envelope. Does this affect the conclusion that in most cases the baseline population size is not attained after the downturn?

As shown in this figure, Table 1, and the Methods, b (for baseline) is the value of the SPD at the start of a downturn, which per our criteria (Line 54-55) is when a downturn commences relative to the fitted null model. The parameter e (for endpoint) is where the downturn intersects with the critical envelope again and can likewise no longer be considered a downturn in statistical terms. We do not think it is warranted to select arbitrary points *outside the scope of the downturns themselves* as points of reference. Adopting this approach would necessarily involve determining pre-downturn SPD peaks on a case-by-case basis and, in our view, diminish the comparability of our results.

In response to the point, therefore, we broadly agree with the reviewer's general point that a *disturbance* (in the sense of our definition in Line 54-55) can begin to have an effect before a *downturn* resolves in radiocarbon time-frequency data. This would impact on our conclusion that baseline population size is not attained after a downturn, since the point of reference would be different from the start of the downturn. Speculatively, we suggest that baseline relative population size would *virtually never* be attained, since the point of reference would likely be much higher than the parameter b , strengthening our assertion that baselines are rarely attained by the end of downturns.

Nevertheless, we wish to emphasise that our focus is on the downturns themselves as a proxy for relative archaeological population/activity, rather than using different, extraneous disturbance events as points of reference. These, where they are evident, often possess considerable uncertainty themselves. In cases where they are not evident, it would be challenging to reliably select baselines without making arbitrary decisions about what constitutes an event's 'start'. Metaphorically speaking, the analysis functions on periods where disturbances 'bite' (resolved as downturns) rather than when they may or may not be ultimately set in motion.

4) Lines 103-104: The authors demonstrate that mobility is the second most common type of disturbance. Mobility can influence changes in the SPD curve in the absence of changes in population size. If the population switches from a more mobile to a less mobile way of living, this will be reflected as a downturn in the SPD curve. Is this kind of effect taken into consideration?

We observe that the effect the reviewer describes is not necessarily straightforward. While higher mobility might lead to more sites and ultimately more dates as they point out, less mobile settlement systems may lead to more densely occupied sites, which not only generate more datable material but are also more likely to be detected and recovered, and hence dated. We have employed best practices in attempting to account for such sampling biases in our data, by binning and thinning our samples as described in the supplementary material.

Moreover, the Mobility parameter itself aims to capture downturns in which regional and/or settlement abandonment are evident in the associated archaeological records, employing a relatively neutral term rather than one with necessarily negative connotations such as "abandonment". The Carrying Capacity parameter also uses this as a criterion, however, we elected to include heightened mobility in response to environmental drivers as a distinct type of downturn within the 'Mixed' category to distinguish this from 'purely' Cultural drivers (in the sense of Table 1). In a broad sense and where supported by expert interpretation of archaeological information (by co-authors ER for Wyoming/Utah, SS for Norway), the parameter 'Mobility' does account for the effect the reviewer highlights.

5) If I understood correctly, for each downturn event, resilience and resistance are measured based on the SPD curve section associated with the event, and the cumulative frequency of downturns up to a particular downturn is associated with the focal event. If this is correct (it seems to make most

sense and this is what entries in Table S2 suggest), this should be more clearly stated, as it is easy to misunderstand that the FD is calculated at the regional level which would result in only 16 possible FD values.

The reviewer's interpretation is correct. We have clarified how FD is calculated in the Methods, which is on a per-event, rather than a per-region basis as the reviewer states.

6) Line 378 and line 384. A figure number is missing.

Corrected to Figure S1 and Figures S2-3, respectively.

7) Identification of downturns. I saw in the supplementary files that the `resmet()` function is mentioned which performs such tests on the SPD curve to identify significant downturns. I did not quite understand how is this test different from the Edinborough et al. (2017) test? All this should be clarified as the number of downturn events depends on the method of their identification. Moreover, downturns can happen in time intervals when the entire SPD curve is in the "red zone", above the upper 95% envelope limit. Why are these not taken into account?

The reviewer rightly requests that we clarify how the `resmet()` function works: the Methods section has been amended to improve clarity and provide more detail.

In specific response to this comment, the test works as summarised by the equations in Table 1, sketched in Figure 1B, and the accompanying code in the file `mcmc.R`. The equations necessitate a start-, mid-, and end-point in order to calculate the resistance-resilience metrics, following Orwin and Wardle (2004). The code, as written, automatically detects the downturns that resolve in objects of class 'spdppc' produced by the `PostPredSPD()` function in `nimbleCarbon`, to obtain the values of the empirical SPD at these three points in time for use in the equations. The function outputs a data frame containing the value of both metrics, as well as the duration, end- and start-times of downturns, and the time to SPD minimum, all in calendar years Before Present. Column 'LD' (shorthand for lag/duration) is the Time to SPD minimum normalised by the downturn duration, which in the manuscript we have termed 'Pace'.

The `p2pTest` (per Edinborough et al. 2017) in contrast works on *user-specified* intervals and does not directly consider whether a downturn is present or not, although it does carry out a significance test for differences between the user-set SPD values. Our function does not require this, as by definition downturns are statistically significant events relative to the fitted null model. In other words, the `p2pTest()` function appears to employ arbitrary points in time. The `resmet()` function targets intervals on an SPD that are known *a priori* to be of interest. The `p2pTest()` function directly compares SPD values, whereas the `resmet()` function derives meaningfully comparable metrics from SPD values. Theoretically, one could develop a function that combines these features, however, `rcarbon` is not employed for statistical modelling in our workflow, simply calibration and aggregation of radiocarbon dates; the `resmet()` function was developed with the functionality of `nimbleCarbon` in mind.

Regarding the second part, the reviewer points out that SPDs can dip precipitously while still remaining above the critical envelope drawn from the fitted model. While true, such phases are not in the strictest sense to be understood as downturns, since the overall, aggregated value of time-frequency radiocarbon data represented by the SPD is in excess of our fitted models. The MCMC modelling diagnostics (located in 'output/mcmc_diagnostics' in the supplementary information) universally indicate, through evidence of adequate mixing and chain convergence, that exponential

null models are an adequate fit to data, produce comparable outputs, and are advantageously simple (3 parameters) over multi-component models or models that require additional assumptions on parameters such as carrying capacity. Therefore, periods highlighted by the reviewer as “downturns” (where an SPD may drop yet remain above or within the critical envelopes) are empirically *not* downturns per our working definition. In a theoretical sense, an observed dip can be caused by sampling error or calibration effect, however, those outside the envelope are more likely to be genuine downturns. Of course, there might be some false negatives (due to sample sizes or small effect sizes), but false positives are far less likely on this basis, meaning if anything that the number of downturns detected is an underestimate.

In the future, we hope to extend the functionality of `resmet()` to work on growth rates of SPDs rather than absolute SPD values, which would likely highlight phases of the kind the reviewer points out as “downturns” *in the growth rate*. We anticipate this would require extension of the `nimbleCarbon` package itself rather than writing an external function such as `resmet()`, as `nimbleCarbon` does not presently possess the functionality to produce rate-of-change curves from posterior predictive checks. For these reasons, as well as the fact that the original studies we draw on used SPDs rather than their growth rates, the rate of change of the SPD itself is not the target of our analyses in this instance.

8) The main text states that the total number of downturns is 154, yet in Table S2 and the supplementary CSV file, there are 169 downturns. What is the reason for this mismatch?

The supplementary information contains *all* downturns ($n = 169$) produced by our Bayesian MCMC modelling of summed probability distributions. The supplementary code (`statisticalmodelling.R`, Line 17) removes all events that are shorter than 10 years in duration, resulting in 154 downturns for use in the analysis.

Our justification for removing these extremely short downturns is that they are likely to be artefacts of the posterior predictive check carried out on the fitted exponential models. They largely tend to appear very shortly before or after longer periods of downturn (which are retained for the statistical analysis). However, we do not feel there is any justification for “lumping” the shorter downturns into longer ones. This would effectively be treating periods where no population downturns are evident (i.e. the gaps between closely adjacent events that are ≤ 10 and > 10 years long) *as if* there were a downturn there. It is our judgement that it is preferable to remove these events.

In a small number of other cases, these short downturns appear standalone during periods where the material and/or written records have an absence of support for negative population events. Although we have retained *longer* events for which there is little support for causes or drivers, often due to sparse research (logged as `Unclear/U` in the relevant data columns), in light of the above, it is likely that these too are artefacts of analysis and so they too are removed from the formal statistical modelling. We have clarified these points in the Methods.

9) I had some troubles with running the code for the Norway example. When I ran the lines from the "Run model" section, the following error message appeared:

```
Error in checkForRemoteErrors(val) :
```

```
one node produced an error: Failed to create the shared library. Run 'printErrors()' to see the compilation errors.
```

Yet when I ran the `printErrors` command, the message was that there were no errors. But I could not generate the output.

We have unsuccessfully attempted to reproduce this error on four workstations, including that of the main developer of nimbleCarbon. In our search, we came across a thread on the nimble mailing list that describes a similar error: <https://groups.google.com/u/1/g/nimble-users/c/XgD97zSip9o/m/FQydfgv-BQAJ>

Although there is no solution as such, the discussion indicates that the error is somehow down to the way nimble (the package underlying nimbleCarbon) runs models in parallel across cores.

As a workaround, the reviewer may wish to attempt the analysis on a single core per the package vignette: https://cran.r-project.org/web/packages/nimbleCarbon/vignettes/nimble_carbon_vignette.html. This might avoid the issue and allow the code to run, albeit extremely slowly.

However, as noted, we cannot reproduce this error on multiple different devices and operating systems. Without precise information on the nature of the reviewer's hardware, software versions, etc., we cannot provide an exact solution. In all cases, we were able to obtain fitted models from the code and data that already accompanies the manuscript, and we anticipate that the error reported by the reviewer will only be experienced by a minority of users. Even assuming this, single-core runs of the models should enable full reproducibility. Additionally, all of our MCMC outputs are in the supplementary information (following the convention `params_*.rdata`), meaning that posterior predictive checks and metrics can still be obtained for verification.

Reviewer Reports on the First Revision:

Referees' comments:

Referee #1 (Remarks to the Author):

I am happy with the revised version of the manuscript and the authors' replies to comments. All points raised in the previous round of review have been addressed with good arguments. I congratulate the authors on an excellent research and I suggest the editor to accept the manuscript for publication.

Referee #2 (Remarks to the Author):

The authors have taken account and responded appropriately to all major points raised in my earlier critical remarks and assessment.

They appear also to have responded appropriately to the more detailed critical questions raised by reviewer #1 regarding the statistical analysis, although this is not an area in which I have specialist knowledge.

In my opinion this article should now be published.